

# Oceanic forcing of the Eurasian Ice Sheet on millennial time scales during the Last Glacial Period

Jorge Alvarez-Solas[1,2], Rubén Banderas[1,2], Alexander Robinson[1,2,3,4], and Marisa Montoya[1,2]

[1]Dpto. Astrofísica y Ciencias de la Atmósfera; Facultad de Ciencias Físicas; Universidad Complutense de Madrid (UCM)
[2]Instituto de Geociencias (UCM-CSIC), Madrid, Spain
[3]Potsdam Institute for Climate Impact Research (PIK), Potsdam, Germany
[4]Faculty of Geology and Geoenvironment, National and Kapodistrian University of Athens, Greece

*Correspondence to:* Jorge Alvarez-Solas (jorge.alvarez.solas@fis.ucm.es)

**Abstract.**

The last glacial period (LGP; ca.110-10 ka BP) was marked by the existence of two types of abrupt climatic changes, Dansgaard-Oeschger (D/O) and Heinrich (H) events. Although the mechanisms behind these are not fully understood, it is generally accepted that the presence of ice sheets played an important role in their occurrence. While an important effort has been made to investigate the dynamics and evolution of the Laurentide Ice Sheet (LIS) during this period, the Eurasian Ice Sheet (EIS) has not received much attention, in particular from a modeling perspective. However, meltwater discharge from this and other ice sheets surrounding the Nordic Seas is often implied as a potential cause of ocean instabilities that lead to glacial abrupt climate changes. Thus, a better understanding of its variations during the LGP is important to understand its role in glacial abrupt climate changes. Here we investigate the response of the EIS to millennial-scale climate variability during the LGP. We use a hybrid, three-dimensional, thermomechanical ice-sheet model that includes ice shelves and ice streams. The model is forced offline through a novel perturbative approach that includes the effect of both atmospheric and oceanic variations and provides a more realistic treatment of millennial-scale climatic variability than conventional methods. Our results show that the EIS responds with enhanced iceberg discharges in phase with interstadial warming in the North Atlantic. Separating the atmospheric and oceanic effects demonstrates the major role of the ocean in controlling the dynamics of the EIS on millennial time scales. While the atmospheric forcing alone is only able to produce modest iceberg discharges, warming of oceanic surface waters leads to much higher rates of iceberg discharges as a result of relatively strong basal melting within the margins of the ice sheet. Together with previous work, our results provide a consistent explanation for the timing of the responses of the LIS and the EIS to glacial abrupt climate changes.

## 1 Introduction

The last glacial period (LGP; ca.110-10 ka before present, BP) was marked by the existence of two types of abrupt climatic changes: Dansgaard-Oeschger (D/O) and Heinrich (H) events (e.g. Alley et al., 1999). D/O events are identified in Greenland ice-core records as regional abrupt warmings by up to 16°C (Huber et al., 2006; Kindler et al., 2014) from cold (stadial) to relatively warm (interstadial) conditions within decades (Dansgaard et al., 1993) followed by a gradual cooling interval

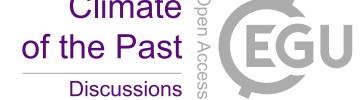

lasting from centuries to millennia and an ultimate phase of rapid cooling back to stadial conditions (Steffensen et al., 2008). Superimposed on the millennial-scale variability associated with D/O events, an additional lower-frequency climatic cycle is identified. So-called Bond cycles are flanked by prolonged stadials ending with prominent D/O events within about 7-10 kyr (Bond et al., 1993). Preceding these, and concomitant with the culmination of the prolonged stadials, H events are registered

in North Atlantic marine sediments as layers of remarkably high concentrations of ice-rafted debris (IRD) (Heinrich, 1988) as a result of massive iceberg discharges from the Laurentide ice-sheet (LIS) (Hemming, 2004).

While significant effort has been invested in understanding the role of the LIS in glacial abrupt climate changes, the dynamics of the Eurasian Ice Sheet (EIS) during the LGP has received comparatively less attention from a modeling perspective. However, improving our understanding of its evolution and response to past climate changes is important for a number of

reasons. First, constraining freshwater inputs into the North Atlantic Ocean is crucial for a better understanding of the driving mechanisms of glacial abrupt climate changes (Rasmussen and Thomsen, 2013), since meltwater discharge from the ice sheets surrounding the Nordic Seas is often implied as a cause of ocean instabilities. Precursor events could possibly have originated from the European and Icelandic ice sheets (Grousset et al., 2000; Scourse et al., 2000). Meltwater peaks in the Norwegian Sea during Marine Isotopic Stage 3 (MIS 3) have been associated with H events and millennial-scale climate variability (Lekens

et al., 2006). From a broader perspective, the EIS, consisting of the Fennoscandian, the British Isles and the Barents-Kara ice sheets (FIS, BIIS and BKSIS, respectively) contained a large marine-based sector at its maximum extension (Hughes et al., 2016) that was exposed to oceanic variations, and the BKSIS is often considered as an analog for the current West Antarctic ice sheet (WAIS). At the LGM both had a similar size, but while the WAIS endured the deglaciation, the BKSIS completely disappeared (Andreassen and Winsborrow, 2009). Understanding the underlying mechanisms would provide important insights

into the future evolution of the WAIS (Gudlaugsson et al., 2013, 2017).

Reconstructing the EIS response to past glacial abrupt climate changes prior to the LGM has been difficult, in part because, in reaching its maximum extent, the ice sheet eroded and removed nearly all older deposits. Nevertheless, the available paleodata indicate that during MIS 3 the EIS was highly dynamic, with its advance and retreat closely linked to stadials and interstadials. In this line, records from Norway (Mangerud et al., 2003, 2010; Olsen et al., 2002), Finland (Helmens and Engels, 2010) and

Sweden (Wohlfarth, 2010) indicate rapid and rythmic ice-sheet variations in western Scandinavia, with advances and retreats during stadials and interstadials, respectively. Recent records also indicate enhanced meltwater discharges during interstadials from the Svalbard-Barents Sea ice sheet and probably also from the Scandinavian ice sheet (Rasmussen and Thomsen, 2013). The resolution and quality of geophysical data across marine sectors has improved considerably in the past decade (Hughes et al. (2016) and references therein). The results confirm substantial variations of the EIS volume, with the largest uncertainties

in marine sectors of the ice sheets. Strong variations in the deposition of IRD suggests high co-variability of the BIIS with changes in ocean sea surface temperature (Hall et al., 2011; Scourse et al., 2009) and variations in EIS ice streams (Becker et al., 2017). North Atlantic marine sediment records register widespread variations of IRD input throughout the LGP indicating variations of iceberg rafting from virtually all surrounding ice sheets. Sources and timing differ among different sites. A dominant periodicity equal to that of D/O events was identified in the Irminger Sea, with the largest IRD peaks at the end of

stadials originating in the Iceland and Greenland ice sheets (von Kreveld et al., 2000). Strong millennial-scale iceberg rafting



variability of the BIIS has been documented as well in the North Sea (Hall et al., 2011; Peck et al., 2007; Scourse et al., 2009), but enhanced IRD seems to occur both during interstadials and during stadial periods. For the FIS, IRD records in the Norwegian Sea show the characteristic D/O periodicity, with IRD occurring just before interstadial transitions (Lekens et al., 2006). More recently, however, an increase in IRDs from Fennoscandia during interstadials has been reported (Dokken et al.,
2013; Becker et al., 2017).

Progress has been achieved also in the past decade using ice-sheet models. Siegert and Dowdeswell (2004) used inverse modelling to simulate the EIS evolution during the second part of the LGP, matching the geological evidence presented by optimizing the fit with data. Forsström and Greve (2004) used subsequent versions of a three-dimensional, polythermal ice-sheet model to simulate the EIS evolution throughout the LGP. Important variations in the EIS ice volume in response to
temperature and precipitation variations were simulated. Clason et al. (2014) additionally included a parameterisation of surface meltwater enhanced sliding. In both cases too much ice was simulated in the northeastern EIS. Gudlaugsson et al. (2017) used the same model but introducing a simple representation of the subglacial hydrological system, focusing on its role in the temporal evolution of the EIS. Recently, an ice-sheet model constrained by data has been used to simulate the EIS evolution throughout part of LGP (Patton et al., 2016). The model targets the most probable EIS distribution at different time slices
and reproduces substantial ice-volume variations. However, all of these models suffer from limitations, such as the use of the shallow-ice approximation (SIA) and its associated lack of an explicit treatment of the oceanic forcing. Marshall and Koutnik (2006) investigated the production of icebergs from all the North American ice sheets with a parameterized calving model. They found different behaviors on millennial time-scales depending on the local glaciological and climatic characteristic, with increased iceberg production both during stadials (e.g. from Iceland) or during interstadials (e.g. from Barents Sea). Nonethe-
less, sub-marine melting at the grounding line has not been explicitly considered until now and its impacts on millennial-scale variability have not been investigated up to now from a modelling perspective.

Here, we investigate the response of the EIS to millennial-scale climate variability during MIS 3 using a three-dimensional ice-sheet model. To this end, a novel offline approach is used that provides a better representation of millennial-scale climate variability (Banderas et al., 2017). In addition, for the first time, both the atmospheric and oceanic effects of millennial scale
climate variability associated with glacial abrupt climate changes are considered. This facilitates the quantification of the relative contribution of surface (ablation) and dynamic processes related to ice-ocean interactions.

The paper is organized as follows: in Section 2 the ice-sheet model, the forcing method and the experimental setup are described. In Section 3 the response of the EIS to the imposed forcing is shown, the focus being the evolution of its ice volume, its impact on sea level and the mechanisms behind meltwater and ice discharge. Finally, the main conclusions are summarised
in Section 5.





## 2 Model and experimental setup

### 2.1 Model

The model used in this study is the ice-sheet model GRISLI-UCM, an extension of the original model GRISLI developed by Ritz et al. (2001). GRISLI-UCM is a hybrid three-dimensional thermomechanical ice-sheet model. Inland flows through deformation under the Shallow Ice Approximation (SIA, Hutter, 1983). Ice shelves and ice streams are described following the Shallow Shelf Approximation (SSA, MacAyeal, 1989). Ice streams (areas of fast flow, typically faster than $10^2$ $m$ $a^{-1}$) are considered as dragging ice shelves, allowing basal movement of the ice (Bueler and Brown, 2009). Basal stress under ice streams is proportional to ice velocity and to the effective pressure of ice. The locations of the ice streams are determined by the presence of basal water within areas where the sediment layer is saturated. GRISLI-UCM thus explicitly calculates grounding line migration, ice-stream and ice-shelf velocities. This allows the model to properly represent both grounded and floating ice. GRISLI-UCM uses finite differences on a staggered Cartesian grid at a 40 km resolution, corresponding to 224×208 grid points for the Northern Hemisphere domain, including the EIS, with 21 vertical levels. By default, initial topographic conditions are provided by surface and bedrock elevations built from the ETOPO1 dataset (Amante and Eakins, 2009) and ice thickness (Bamber et al., 2001). The surface mass balance is given by the sum of accumulation and ablation, both of which are calculated from monthly surface air temperatures (SATs) and monthly total precipitation. Accumulation is calculated by assuming that the fraction of solid precipitation is proportional to the fraction of the year with mean daily temperature below 2°C. The daily temperature is computed from monthly SATs assuming that the annual temperature cycle follows a cosine function. Ablation is calculated using the positive-degree-day (PDD) method (Reeh, 1989). Basal melting inland depends on pressure and water content at the base of the ice sheet as well as the geothermal heat flux, which is prescribed from a recent reconstruction (Shapiro and Ritzwoller, 2004). Basal melting for floating ice is computed using a linear temperature anomaly with respect to the freezing point. The details of the implementation of the boundary conditions (SMB and oceanic basal melting) in this particular study are given below (Section 2.2).

### 2.2 Offline forcing method

SMB and oceanic basal melting are obtained through a time-varying synthetic climatology built through a novel method that is found to provide a more realistic offline forcing for ice-sheet models than classical offline methods (Banderas et al., 2017). The method follows a perturbative approach in the sense that the forcing combines the present-day climatology, obtained from observational data, together with simulated anomalies. But in contrast to usual offline forcing methods, orbital and millennial scale variabilities are not lumped in a sole anomaly pattern but differentiated. The method thus combines present-day observations, simulated Last Glacial Maximum (LGM) anomalies relative to present, scaled by an orbital-timescale index, and simulated stadial-interstadial anomalies, scaled by a millennial-timescale index:

$$\boldsymbol{T}^{\mathrm{atm}}(t) = \boldsymbol{T}_0^{\mathrm{atm}} + (1 - \alpha^{\star}(t))\,\boldsymbol{\Delta T}_{\mathrm{orb}}^{\mathrm{atm}} + \beta^{\star}(t)\,\boldsymbol{\Delta T}_{\mathrm{mil}}^{\mathrm{atm}} \tag{1}$$

$$\boldsymbol{P}(t) = \boldsymbol{P}_0\left\{\alpha^{\star}(t) + (1 - \alpha^{\star}(t))\,\delta\boldsymbol{P}_{\mathrm{orb}}\left[(1 - \beta^{\star}(t)) + \beta^{\star}(t)\delta\boldsymbol{P}_{\mathrm{mil}}\right]\right\} \tag{2}$$



Here, $\boldsymbol{T}^{\mathrm{atm}}(t)$ and $\boldsymbol{P}(t)$ are the SAT and precipitation fields at time $t$. $\boldsymbol{T}_0^{\mathrm{atm}}$ and $\boldsymbol{P}_0$ are the ERA-INTERIM present-day SAT and precipitation climatologies (Dee et al., 2011). $\boldsymbol{\Delta T}_{\mathrm{orb}}^{\mathrm{atm}} = \boldsymbol{T}_{\mathrm{lgm}}^{\mathrm{atm}} - \boldsymbol{T}_{\mathrm{pd}}^{\mathrm{atm}}$ and $\delta \boldsymbol{P}_{\mathrm{orb}} = \boldsymbol{P}_{\mathrm{lgm}}/\boldsymbol{P}_{\mathrm{pd}}$ are the orbital temperature anomaly and precipitation ratio relative to the present day (Figure 2d-f in Banderas et al., 2017), respectively, obtained from previous equilibrium simulations for the preindustrial and LGM climates performed with the CLIMBER-3$\alpha$ model (Montoya

and Levermann, 2008). $\boldsymbol{\Delta T}_{\mathrm{mil}}^{\mathrm{atm}} = \boldsymbol{T}_{\mathrm{is}}^{\mathrm{atm}} - \boldsymbol{T}_{\mathrm{st}}^{\mathrm{atm}}$ and $\delta \boldsymbol{P}_{\mathrm{mil}} = \boldsymbol{P}_{\mathrm{is}}/\boldsymbol{P}_{\mathrm{st}}$ are the millennial temperature anomaly and precipitation ratio, respectively, for the interstadial relative to the stadial state (Figure 1). Note bold symbols indicate two-dimensional spatial fields. The stadial mode in our study is represented by a climate simulation of the LGM with CLIMBER-3$\alpha$ (Montoya and Levermann, 2008). The interstadial mode is taken from a recent glacial transient simulation performed with the same model under glacial climatic conditions, but with intensified NADW formation (Banderas et al., 2015). $\alpha^\star$ and $\beta^\star$ are two

indices that separately modulate the contribution of the orbital and millennial anomalies. Both were built based on two recent complementary temperature reconstructions over Greenland, one from the NGRIP ice-core record for the LGP (Kindler et al., 2014), and the other one from several ice-core records for the Holocene (Vinther et al., 2009). Their combination (hereafter, the KV reconstruction) results in a continuous temperature reconstruction for Greenland for the past 120 ka (Banderas et al., 2017). $\alpha^\star$ is obtained after applying a low-pass frequency filter ($f_c = 1/18 \, \mathrm{ka}^{-1}$) to the original KV reconstruction based on a spectral

decomposition; $\beta^\star$ is obtained following a similar procedure but retaining the high frequency signal. Both indices are tuned in such a way that the resulting synthetic temperature time series at the NGRIP site exactly matches the KV reconstruction (this distinguishes $\alpha^\star$ and $\beta^\star$ from the raw $\alpha$ and $\beta$ indices previous to this tuning; Banderas et al. (2017)).

Finally, the net basal melting rate for floating parts $\boldsymbol{B}$ is assumed to follow a linear relation (e.g. Beckmann and Goosse (2003)):

$$\boldsymbol{B} = \kappa \left( \boldsymbol{T}^{\mathrm{ocn}} - \boldsymbol{T}_f \right) \tag{3}$$

where $\boldsymbol{T}^{\mathrm{ocn}}$ is the oceanic temperature close to the grounding line, $\boldsymbol{T}_f$ is the temperature at the ice base, assumed to be at the freezing point, and $\kappa$ is the heat flux exchange coefficient between ocean water and ice at the ice-ocean interface. Following the approach described above, $\boldsymbol{T}^{\mathrm{ocn}}(t)$ is given by an expression analogous to Eq. 1. Thus Eq. 3 can be rewritten as:

$$\boldsymbol{B} = \boldsymbol{B}_0 + \kappa \left[ (1 - \alpha^\star(t)) \, \boldsymbol{\Delta T}_{\mathrm{orb}}^{\mathrm{ocn}} + \beta^\star(t) \, \boldsymbol{\Delta T}_{\mathrm{mil}}^{\mathrm{ocn}} \right] \tag{4}$$

where $\boldsymbol{B}_0 = \kappa(\boldsymbol{T}_0^{\mathrm{ocn}} - \boldsymbol{T}_f)$ represents the present-day oceanic basal melting rate.

## 2.3   Experimental setup

We herein investigate the response of the EIS to millennial-scale climate variability during MIS 3. The starting point of our experiments is a control-run ice-sheet simulation with constant bounday conditions for MIS 3 that provides a representative configuration of the EIS for that time period (hereafter, CTRL; Figure 1a, b). To this end, $\alpha^\star$ was set to its value at 40 ka BP,

that is, $\alpha^\star = \alpha_{40K}^\star = -0.1$ and $\beta^\star = 0$ to preclude millennial-scale variations. Note however these values are to a certain extent arbitrary; they are intended to provide a stable mean background state similar but not necccessarily identical to background



MIS 3 conditions. Thus:

$$\boldsymbol{T}_{40K}^{\mathrm{atm}} = \boldsymbol{T}_0^{\mathrm{atm}} + (1 - \alpha_{40K}^\star)\,\boldsymbol{\Delta T}_{\mathrm{orb}}^{\mathrm{atm}} \tag{5}$$

$$\boldsymbol{P}_{40K} = \boldsymbol{P}_0\left[\alpha_{40K}^\star + (1 - \alpha_{40K}^\star)\,\delta \boldsymbol{P}_{\mathrm{orb}}\right] \tag{6}$$

$$\boldsymbol{B}_{40K} = \boldsymbol{B}_0 + \kappa\,(1 - \alpha_{40K}^\star)\,\boldsymbol{\Delta T}_{\mathrm{orb}}^{\mathrm{ocn}} \tag{7}$$

Note that although Eq. 7 is formally correct and consistent with the scheme used, in contrast to the present-day SAT or precipitation the present-day rate of oceanic basal melting cannot be determined. Thus, in practice we replace this equation by directly tuning the value of $\boldsymbol{B}_{40K}$ to obtain a reasonable ice-sheet configuration at 40 ka BP given the atmospheric forcing fields expressed by equations 5-6. To this end, a constant basal melting rate of 0.1 m a$^{-1}$ is assumed. This is found to allow the growth of European ice-sheets to an extent that satisfactorily agrees with previous reconstructions (Svendsen et al., 2004;
Kleman et al., 2013).

Our forcing method allows to investigate the response of the EIS solely to millennial-scale climate variability at MIS 3 by keeping constant the orbital component of the forcing ($\alpha^\star = \alpha_{40K}^\star$) and letting $\beta^\star$ vary throughout the LGP (eqs. 1, 2 and 4). In order to assess the relative roles of the atmosphere and the ocean, three independent experiments have been carried out. First, an atmospheric-only forced simulation (ATM) in which the time evolution of SAT and precipitation on millennial time scales
is considered, while the oceanic forcing is kept constant to MIS 3 (i.e., 40 ka BP) background climatic conditions. Thus:

$$\boldsymbol{T}^{\mathrm{atm}}(t) = \boldsymbol{T}_{40K}^{\mathrm{atm}} + \beta^\star(t)\,\boldsymbol{\Delta T}_{\mathrm{mil}}^{\mathrm{atm}} \tag{8}$$

$$\boldsymbol{P}(t) = \boldsymbol{P}_{40K}\left[(1 - \beta^\star(t)) + \beta^\star(t)\,\delta \boldsymbol{P}_{\mathrm{mil}}\right] \tag{9}$$

$$\boldsymbol{B}(t) = \boldsymbol{B}_{40K} \tag{10}$$

Second, an oceanic-only forced simulation OCN in which the atmospheric forcing is kept constant while the oceanic basal
melting is allowed to vary at millennial timescales around its background MIS 3 value:

$$\boldsymbol{T}^{\mathrm{atm}}(t) = \boldsymbol{T}_{40K}^{\mathrm{atm}} \tag{11}$$

$$\boldsymbol{P}(t) = \boldsymbol{P}_{40K} \tag{12}$$

$$\boldsymbol{B}(t) = \boldsymbol{B}_{40K} + \kappa\,\beta^\star(t)\,\boldsymbol{\Delta T}_{\mathrm{mil}}^{\mathrm{ocn}} \tag{13}$$

The magnitude and sign of oceanic temperature anomalies $\boldsymbol{\Delta T}^{\mathrm{ocn}}$ depends on the depth at which $\boldsymbol{T}^{\mathrm{ocn}}$ is considered (Figure 1e,
f). The latter represents the oceanic temperature at the base of the ice shelf. In our simulations, a large part of the NE sector of the EIS is marine based, with depths below 500 m at the margins. Thus, we herein identify $\boldsymbol{T}^{\mathrm{ocn}}$ with the sea surface temperature (SST). Note that this differs from previous studies focusing on the response of the LIS to ocean circulation changes, in which the existence of large and thick ice shelves led to the consideration mainly of subsurface oceanic temperatures (e.g. Alvarez-Solas et al. (2013)). In this case, an ensemble of simulations for different values of $\kappa$ have been considered to evaluate the
sensitivity of the EIS to the forcing.

Finally, a simulation ALL combining both the atmospheric and the oceanic forcings:



$$\boldsymbol{T}^{\mathrm{atm}}(t) = \boldsymbol{T}^{\mathrm{atm}}_{40K} + \beta^{\star}(t)\,\boldsymbol{\Delta T}^{atm}_{\mathrm{mil}} \tag{14}$$

$$\boldsymbol{P}(t) = \boldsymbol{P}_{40K}\left[(1-\beta^{\star}(t)\,) + \beta^{\star}(t)\,\delta\boldsymbol{P}_{\mathrm{mil}}\right] \tag{15}$$

$$\boldsymbol{B}(t) = \boldsymbol{B}_{40K} + \kappa\,\beta^{\star}(t)\,\boldsymbol{\Delta T}^{\mathrm{ocn}}_{\mathrm{mil}} \tag{16}$$

In all experiments $\beta^{\star}(t)$ dictates the millennial-scale variability of the forcings (Figure 2). Because our simulated stadial-to-
interstadial transition results from an intensification of the AMOC, positive $\beta^{\star}$ values imply an increase in $\boldsymbol{T}^{\mathrm{atm}}$ relative to
its background MIS 3 value (e.g., Eq. 14 and Figure 1). As a consequence, the atmosphere warms at interstadials relative to
stadial periods, as reflected by the $\boldsymbol{\Delta T}^{\mathrm{atm}}_{\mathrm{mil}}$ millennial-scale anomaly field (Figure 1c).

## 3   Results

Substantial differences are found in the response of the EIS to the forcing scenarios (Figure 2). Under constant forcing, the
CTRL run shows muted calving on the order of 0.01 Sv (Figure 2b), resulting in negligible millennial-scale sea-level equivalent
(SLE) variations, although a lower frequency SLE fluctuation is found related to internal ice-sheet variability (Figure 2a). In
ATM, the atmospheric forcing alone causes a sequence of calving episodes of reduced magnitude (up to 0.02 Sv) resulting
in modest ice volume variations (up to 2 m SLE) during the most prominent stadial-interstadial transitions. In contrast, the
oceanic forcing in OCN induces pronounced changes in the dynamics of the EIS on millennial time scales, with episodes of
enhanced calving occurring during interstadials. Finally, the combination of both atmospheric and oceanic forcings results in
a similar response of the EIS to that obtained in OCN (Figure 2) as a consequence of the much larger effect of the oceanic
forcing in OCN.

The response of the EIS in the forced experiments has been analyzed in terms of its SMB and ice dynamics. In ATM the EIS
shows a muted dynamic response to the forcing with a stable regime of ice-stream velocities throughout MIS 3 (Figure 2b).
Only in the southwestern (SW) part of the EIS is the atmospheric forcing capable of generating an important reduction in the
EIS ice volume on multi-millennial time scales in response to stadial-interstadial transitions (Figures 3a-b). This is a result of
the spatial pattern of the forcing, with the largest SAT anomalies located around the Nordic seas (Figure 1), thus close to the
SW margin of the EIS (Banderas et al., 2017). Therefore, the ice volume reduction in the SW part of the EIS in ATM is due to
the positive SAT anomaly in this region associated with stadial to interstadial transitions, which leads to enhanced ablation in
the SW part of the EIS (Figure 4). Accumulation barely changes (not shown). Thus, the SMB is strongly reduced in response
to warmer SATS. In turn, reduced SATs during stadials allow the regrowth of the ice sheet up to the continental margin of the
Nordic seas.

In OCN, the EIS reacts to every abrupt surface warming with a substantial ice-flow acceleration in the BSIS, especially in
the Bjørnøyrenna ice-stream (Figures 2b and 3). In addition, synchronous episodes of large iceberg discharges up to ca. 0.1 Sv
from the BSIS leading to large SLE variations of up to 6 m are observed. The more active dynamic response of the EIS in OCN
can be attributed to the increase in SSTs by 2-4°C within the margins of the ice-sheet during stadial to interstadial transitions





(Figure 1) that translates into enhanced basal melting in the margins of the EIS. The SW sector of the EIS also responds to the warmer SSTs, actually with a larger reduction of ice volume than in ATM (Figure 3a-d). However, the main contribution to sea level comes from the northeastern (NE) part of the ice sheet (Figure 2a, dashed and solid blue lines). In this area, higher melting rates lead to a substantial retreat of the grounding line in the BSIS (Figure 3b-c). As a result, acceleration of the Bjørnøyrenna ice stream follows, favoring massive iceberg purges from inland during the interstadial warming (Figures 2 and 3).

Inspection of the temporal evolution of the grounding line position in OCN confirms that ice dynamics control ice-volume variations in the BSIS as opposed to the SMB processes involved in ATM. The migration of the grounding line through time has been characterized by means of an index ($\mu$) that weighs the proportion of non-grounded points in the region of the Barents Sea:

$$\mu(t) = \left(1 - \frac{N_g(t)}{N}\right) \cdot 100 \tag{17}$$

where $N_g(t)$ represents the evolution of the number of points of grounded ice within a fixed area of $N$ points in the Barents Sea region. While in ATM $\mu$ barely changes (Figure 5a, b), in OCN it reflects a synchronous evolution of the grounding line position and the oceanic forcing, with major retreats coinciding with interstadial states (Figure 5c). The direct coupling between the oceanic forcing and the response of the BSIS is also evident from the relatively high negative correlation ($r \simeq -0.70$) found between $\mu$ and ice thickness (Figure 5d). In essence, in response to the grounding-line retreat, acceleration of the flow takes place upstream in the Bjørnøyrenna ice stream, as reflected by the nearly linear positive correlation ($r \simeq 0.93$) found between $\mu$ and velocities in the channel (Figure 5d). This leads to an in-phase relationship between warm interstadial states in the North Atlantic and substantial iceberg discharges from the BSIS (Figure 2c). As a consequence of the destabilization of the ice sheet, important ice-volume variations are observed in the NE part of the EIS during millennial-scale climatic transitions, which added to the minor contribution of the SW retreat, result in fluctuations of up to 6 m SLE (Figure 2d).

In order to investigate the sensitivity of the results to the model parameters, four additional OCN simulations have been carried out with different $\kappa$ parameters between 1-9 $m\ a^{-1}\ K^{-1}$, i.e., bracketing our standard case of $\kappa = 5\ m\ a^{-1}\ K^{-1}$. This choice reflects the inferences based on measurements made on Antarctic ice shelves that a variation of 1 K in the effective oceanic temperature changes the melt rate by ca. 10 $m\ a^{-1}$ (Rignot and Jacobs, 2002; Shepherd et al., 2004). A robust response of the EIS is found, with a more reactive EIS response for increasing $\kappa$ values (Figure 6). The combination of both atmospheric and oceanic forcing leads to similar results to those obtained in OCN (Figures 2 and 3). This is a consequence of the different magnitudes and locations of the two forcings. The highest SAT anomalies are centered around the Nordic Seas, thus inducing relatively modest melting of the ice in the SW part of the EIS through ablation. Meanwhile, increased SSTs within the margins of the ice-sheet are able to trigger a significant grounding-line retreat both in the SW and the NE parts of the EIS.

## 4  Discussion

Our results suggest enhanced iceberg calving from the EIS occurred during interstadials. Recently, IRD peaks of Fennoscandian origin reported from a high-resolution marine sediment core from the Norwegian Sea indicate the presence of more



frequent IRD deposition and thus calving during interstadials than during stadials (Dokken et al., 2013). This result has been corroborated in a compilation of new and previously published data (Becker et al., 2017) clearly showing that within MIS 3, the IRD deposition increases within interstadials. The coeval deposition of carbonate-rich, sorted fine sands and near-surface warming suggests the presence of Atlantic water along the margin, and is interpreted by the authors as the effects of winnowing

due to an intensified AMOC during interstadials, in agreement with our interpretation.

Our results also provide a mechanism to explain the pervasive presence of IRD in the North Atlantic during MIS 3, both during stadials and interstadials, and originating both in the LIS and the EIS. During stadials, the simultaneous appearance of IRD across the wider North Atlantic Ocean can be explained through the build-up of subsurface heat in the high-latitude North Atlantic leading to increased iceberg calving in the presence of large, thick ice shelves, together with lower surface

temperatures allowing for wider dispersal of icebergs (Barker et al., 2015). According to our results interstadials could lead to enhanced calving of the EIS through oceanic surface subglacial melting as a result of the warmer surface conditions and relatively shallow grounding line of this ice sheet.

Our experimental setup is not intended to match the paleorecord, but to provide insight into the response of the EIS to millennial-scale variability. The EIS variations simulated here represent the upper-end amplitude of potential responses during

the whole glacial cycle, due to its large size. Extending the study to cover the whole LGP would require the consideration of orbital variability as part of the forcing. In this case, the EIS would likely be smaller during the mildest phase of MIS 3, thus limiting its contact with the ocean and the production of iceberg discharges.

Note that the identification of IRD layers with increased calving through ice-sheet instabilities must be taken with caution, since it is based on several untested assumptions (Clark and Pisias, 2000): (i) delivery of IRD to a specific site is caused solely

by iceberg calving, versus transport by sea-ice; (ii) an increase in IRD represents an increase in the iceberg flux, versus a greater amount of debris incorporated at the base of the ice sheet that delivers the icebergs, or a greater distance of iceberg transport; (iii) the amoun of IRD carried by all the icebergs is similar, assuming therefore a direct relationship between IRD concentration and iceberg flux. However, the former assumptions have not been confirmed and, thus, the calving-IRD relationship might not be so direct. In addition, ocean temperatures affect melting of icebergs and thus their release of IRD. Variations in ocean

temperatures can alter the IRD released by an iceberg at a certain site, causing variations in IRD deposition even for a constant amount of icebergs produced at the source.

Also, our results may depend on the particular SAT and SST anomaly patterns simulated by our climate model, the magnitudes of the resulting forcing, and the initial size of the simulated EIS. Assessing the sensitivity to these features should be the scope of future work, and ilustrates the need for carrying new simulations of both the interstadial and the stadial states with

more sophisticated climate models. Finally, our study lacks bi-directional coupling between the ice sheet, the atmosphere and the ocean. Eventually the goal is to investigate this matter with fully coupled climate-ice sheet models.

Meltwater discharge from the EIS and other ice sheets surrounding the Nordic Seas is often implied as a cause of ocean instabilities. The same would be the case for iceberg discharges. This issue is beyond the scope of this study; its assessment would require investigating the impact of these freshwater perturbations in deep water formation and the AMOC. Again, proper

assessment requires the use of a coupled climate-ice sheet model.



## 5   Conclusions

We have investigated the response of the EIS to millennial-scale climate variability associated with D/O events through a series
of simulations with a three-dimensional, hybrid ice-sheet model that represents inland ice flow under the SIA and floating
ice shelves and ice streams through the SSA. The model makes use of an offline forcing method that separately accounts for
orbital and millennial-scale climate variability during the LGP, improving the representation of the latter (Banderas et al., 2017).
Atmospheric and ocean forcings associated with millennial-scale variability were considered both separately and together.

Separating the effects of atmospheric and oceanic forcing during the glacial period has allowed us to quantify the contribution
of each to EIS variability. Atmospheric forcing during stadial-interstadial transitions has a very modest effect on the ice sheet,
which is a consequence of the largest SMB changes being confined to SW sector of the EIS, where the forcing is strongest.
In contrast, the oceanic forcing has a much larger effect, through changes in the ice dynamics in the NE sector of the EIS.
Ocean surface warming during interstadials is able to induce a retreat of grounded ice in this part of the EIS through dynamic
processes. As a consequence, significant ice-volume variations result during millennial-scale climatic transitions. Added to the
minor contribution of the SW retreat, this results in sea-level changes on the order of several meters. Sensitivity experiments
for different values of the oceanic heat coefficient parameter show that this is a robust response of the model.

Our results thus support the existence of a highly dynamic EIS during the LGP. They suggest an important role of oceanic
melt forcing through changes in the ocean circulation in controlling the ice-stream activity. Together with previous work
(Alvarez-Solas et al., 2013), they imply that oceanic circulation changes and the associated ocean-ice sheet interactions are
able to explain virtually all ice rafting events in the North Atlantic within MIS 3, from the H events of the LIS during stadials
to those of the EIS during interstadials.

*Competing interests.*   The authors declare no competing interests

*Acknowledgements.*   This work was funded by the Spanish Ministerio de Economía y Competitividad (MINECO) through project MOCCA
(Modelling Abrupt Climate Change, Grant CGL2014-59384-R). R. Banderas was funded by a PhD Thesis grant of the Universidad Com-
plutense de Madrid. A. Robinson is funded by the Marie Curie Horizon2020 project CONCLIMA (Grant 703251). Part of the computations
of this work were performed in EOLO, the HPC of Climate Change of the International Campus of Excellence of Moncloa, funded by MECD
and MICINN. This is a contribution to CEI Moncloa.





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





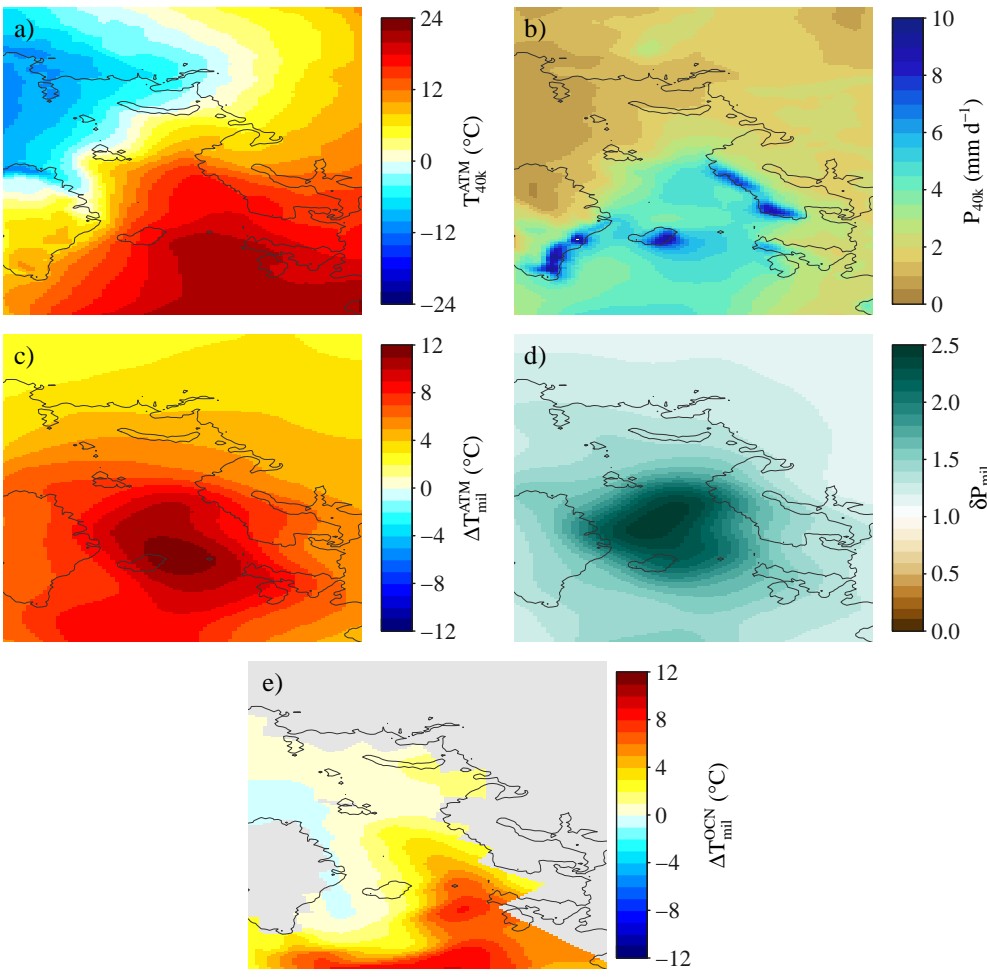

**Figure 1.** Spatial components of the climatic forcing. The upper panels show the atmospheric forcing fields at the MIS 3 (∼40 ka BP) reference state for: **a)** SAT in °C and **b)** precipitation in mm d$^{-1}$. The middle panels show the millennial-scale atmospheric component of the forcing for: **c)** SAT anomalies (interstadial minus stadial) in °C and **d)** precipitation ratio (interstadial to stadial). **e)** Anomalies of SST in °C corresponding to the millennial-scale oceanic component of the forcing.





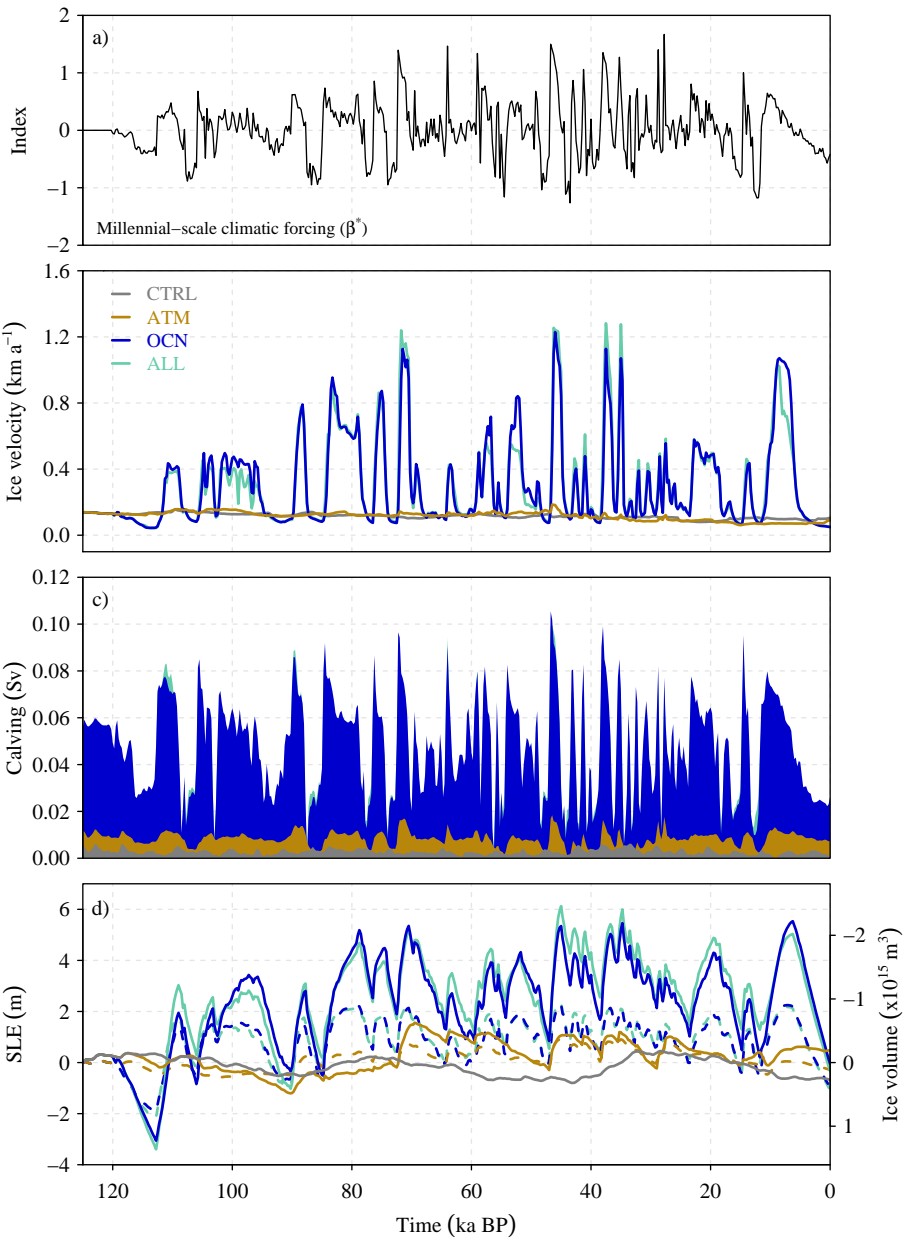

**Figure 2. a)** Temporal component of the millennial-scale climatic forcing ($\beta^\star$ index), and **b)** ice velocities (km a$^{-1}$) in the Bjørnøyrenna ice stream (in the Barents sea ice sheet; see location in Figure 3); **c)** EIS calving (Sv) and **d)** EIS sea-level equivalent (m) related to ice volume variations (m$^3$) with respect to initial conditions for the CTRL (gray) run and for the ATM (gold), OCN (blue), and ALL (turquoise) forcing experiments.





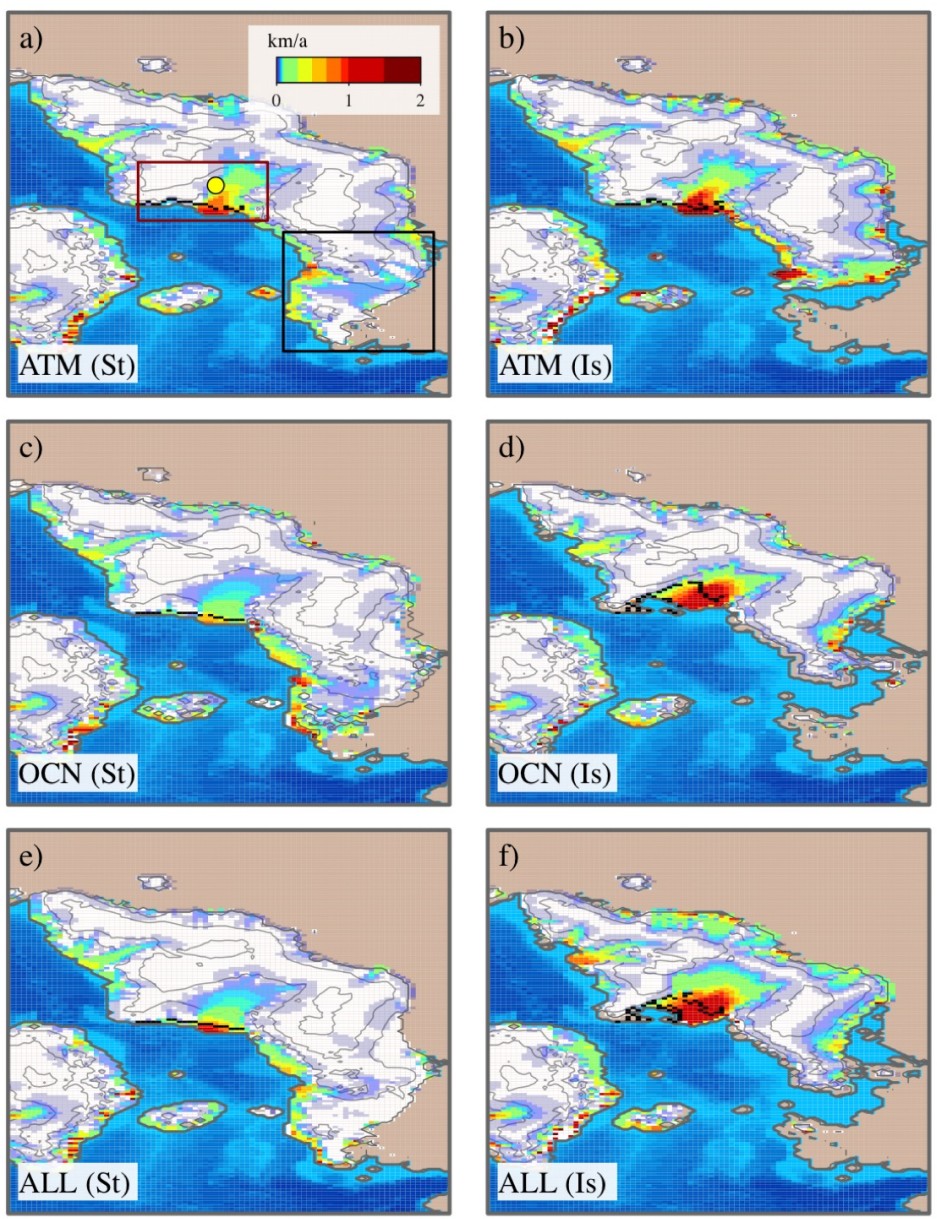

**Figure 3.** Simulated EIS during stadials (left panels) and interstadials (right panels) for the experiments: **a-b)** ATM; **c-d)** OCN and **e-f)** ALL. Shaded colors show ice velocities (km a$^{-1}$). Black thick lines represent the position of the grounding line. Ice thickness is plotted every 750 m as contoured lines. Black and red rectangles in panel **a)** delimit the SW part of the EIS and the NE region where the grounding line retreat has been characterized, respectively.





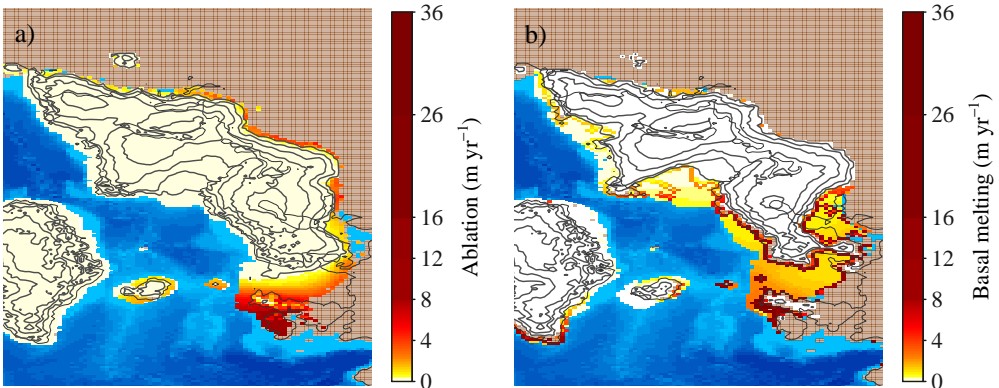

**Figure 4.** Interstadial minus stadial anomalies of: **a)** ablation rate (m $yr^{-1}$) for the ATM experiment and **b)** basal melting rate (m $yr^{-1}$) for the OCN experiment. Ice thickness is plotted every 500 m as contoured lines at the interstadial state for both experiments.



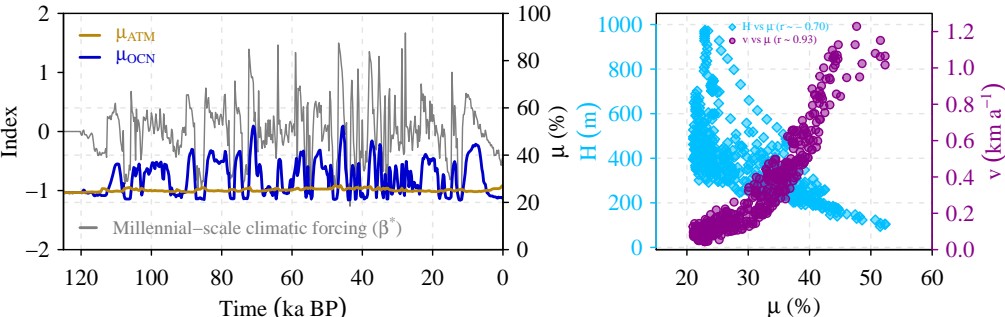

**Figure 5.** Dynamic behavior of the EIS during millennial-scale climatic transitions for the OCN experiment. **a)** Displacement of the grounding line (blue) in the Barents sea ice sheet in response to the climatic $\beta^\star$ forcing (gray). The evolution of the grounding line position has also been shown for ATM (gold). The migration of the grounding line has been characterized as an index $\mu(t)$ that represents the evolution of the number of points of grounded ice $N_g(t)$ over a fixed area of $N$ points in the Barents Sea region : $\mu(t) = 1 - N_g(t)/N$. Positive values of $\mu$ indicate grounding line retreat. **b)** Scatter plot diagram showing the relationship between ice thickness $H$ in the region of the Bjørnøyrenna ice stream and $\mu$ (light blue diamonds; $r(\mu, H) \simeq$ -0.70) as well as the relationship between ice-stream velocities $v$ in the same region and $\mu$ (purple circles; $r(\mu, v) \simeq 0.93$).





**Figure 6.** Response of the EIS to millennial-scale climate variability for the OCN experiment and five different values of the $\kappa$ parameter ($ma^{-1}K^{-1}$). **a)** Temporal component of the millennial-scale climatic forcing ($\beta^{\star}$) index; **b)** grounding line migration index ($\mu$) in the Barents sea ice sheet; **c)** Bjørnøyrenna ice stream velocities (km a$^{-1}$) and **d)** EIS sea-level equivalent (m) related to ice volume variations (m$^3$) with respect to initial conditions. Black solid lines refer to climatic variables obtained from the simulation in which $\kappa = 0.5\ m\ a^{-1}\ K^{-1}$. Black dashed lines represent variables for the CTRL run.