# Peer review of "Oceanic forcing of the Eurasian Ice Sheet on millennial time scales during the Last Glacial Period"

_Climate of the Past, 2017_

## Referee Comment (RC1) · Anonymous Referee #1 · 29 Jan 2018

The study of Alvarez-Solas et al. is well written and of high scientific quality. Using an ice sheet model forced by idealised ocean and atmosphere climates through the last glacial period the authors investigate the response of the European Ice Sheet to millennial scale climate changes. A major finding of the study is that the European Ice Sheet, and in particular fast flowing ice in the vicinity of Bjørnøya trough in the Barents Sea, is highly sensitive to changes in ocean conditions with a minor contribution from changes in atmospheric surface mass balance.

The paper clearly merits publication in climate of the past as it provides important insight into the dynamics of Dansgaard-Oeschger events and the interplay of the ocean

with marine terminating ice sheets of Eurasia. As opposed to previous studies focusing on the dynamics of the Laurentide Ice Sheet this work is novel in providing clues to the contrasting role of the Eurasian ice sheet to these millennial scale climate events of the last glacial. This said, there are a few important comments which have to be addressed before the manuscript can be accepted.

GENERAL COMMENTS: The main conclusion of the work is that the response of the Eurasian Ice Sheet (EIS) during the last glacial period is dominated by changes to the ocean forcing in the Nordic Seas and Arctic Ocean. The study includes a sensitivity study testing the tuning factor used in controlling the ocean melting of ice. However, there is no assessment of the atmospheric forcing of the ice sheet as given by the model. In particular, what is the potential impact of different SMB parameterisations, different atmospheric climate realisations, and the potential impact of ablation on sub-glacial and basal and submarine melt (e.g. Bondzio et al., GRL, 2017).

As stated in the manuscript, the authors use the PDD method to calculate surface ablation. However, the validity of the associated tuning parameters of this parameterisation for the LGM and stadial/interstadial climates investigated is not assessed or discussed. This analysis must be included in order to properly assess the relative role of atmospheric and oceanic forcing.

The climate forcing used is not clearly assessed either. E.g. how do the MIS3 and stadial-interstadial climate changes from the model applied (figure 3) correspond to observations?

Also, the authors chose to apply SSTs as ocean forcing, whereas the ice shelves off the EIS reach a considerable depth (authors state 500m) where ocean temperatures will be significantly different from the surface. The authors should therefore assess the impact of using sub-surface ocean temperatures instead of SSTs for their results. Note that in previous studies by the authors (e.g. Alvarez-Solas et al., CP, 2011), the sub-surface ocean temperature is used as a forcing. Although it is postulated in the manuscript that

sub-surface temperature is more correct for Laurentide type ice shelves, this has to be tested and thoroughly discussed.

In the discussion the authors briefly comment on the difference between freshwater and IRD. This needs further detail. It would be an advantage to include a plot of the amount of meltwater released so as to compare it with the calving. I.e. how much discharge is calving versus melting and what are the different timescales of the two components of the mass balance? This is important in assessing the potential impact on ocean circulation and sea ice.

Basal friction

SPECIFIC COMMENTS:

Page 2, Line 34-35: stated here that IRD peaks found at end of stadials (with reference to von Kreveld et al., 2000) - OBS! This requires an extremely well dated chronology or tephra to assess the phasing between ocean and ice cores - i.e. such inferences relating to timing are extremely important and must be documented carefully.

Page 5, Line 1-5: The forcing is composed of a difference between glacial and present day temperature (and Precip etc) with reference to CLIMBER model runs. However, in figures there is a reference to MIS3 (e.g. fig.1). The use of LGM vs MIS3, stadial and interstadial climate states should be better defined. E.e. what is the stadial mode? The same as LGM from CLIMBER? Also, the interstadial model (with intensified NADW) must be documented - why is NADW stronger and what are key differences with LGM/stadial?

Page 5, Line 27-30: The spin-up of the ice sheet model needs to be clearly described. This is essential for the response of the model to changes in climate forcing. What choices were made for basal friction and ice temperature, rheology etc. What was the spin-up procedure used and how does this impact the results and response of the ice sheet? All these aspects are important to document.

Figure 1: What is shown here? Annual mean values? Clearly define MIS3 as well as the stadial-interstadial experiments (see notes above).

Figure 2: The Greenland ice core index (beta) is key to all the model simulations, but is not well described in the manuscript. The index is quite different from published NGRIP d18O and temperature conversions - it should be clearly stated why the authors choose not to use the normalised ice core temperature data? Also, why is the present value of the index nearly the same as that of the stadial/glacial value?

Figure 2: the ATM experiments shown hardly any oscillations. However, previous work by the the authors as well as other studies show binge-purge like oscillations of the ice sheets given a constant forcing. Why is there no self-sustained oscillations in the version of GRISLI applied here?

Figure 2: Define the dashed lines in the figure and give details in the caption.

Figure 3: The two regions are named SW and NE. This should be clearly stated in caption of figure. However, if possible these names should be more descriptive - e.g something relating to Bjørnøyrenna would be more logical. Also define where this feature is on the map. What is yellow circle?

Figure 3: Note that the land topo in figure 1 and 3 are different. Would be better to use one land topo - or comment on why different (ice sheet vs climate model) and potential impact of this on results.

Figure 5: Not necessary to repeat formula from main text in the caption.

Figure 5: the discussion of the relationship between ice sheet height/velocity and foundling line retreat would benefit from including a discussion of changes in the position of the calving front through time. How does this differ from the grounding line and how does it relates to any of the assessed ice parameters?

TECHNICAL COMMENTS: Line 21: Better to use DO-events and H-events or similar nomenclature. D/O is not a standard form.

---

## Referee Comment (RC2) · Anonymous Referee #2 · 2 Feb 2018

Alvarez-Solas et al. investigate the respective role of atmospheric forcing and oceanic induced sub-shelf basal melting (and refreezing?) rates in the variability of the Eurasian ice sheet during the last glacial period. The paper is scientifically very exciting as whilst a fair amount of climate records exist only little is known about glacial ice sheet variability and in particular the role of the ocean in this variability. However, I have serious doubt on the experimental setup and in particular concerning the basal melting perturbation chosen. If the authors really use an oceanic perturbation allowing for refreezing (and at a greater rate than snow accumulation!) the validity of the paper findings can be largely questioned. I suggest that the authors clarify their methodology as I will not support publication of this paper with the OCN experiment as presented.

I would recommend the authors to perform again their OCN and ALL experiment with an alternative basal melting rate perturbation (e.g. based on a ratio as for precipitation or at least with a positive threshold).

General comments

- I am worry about your experimental setup: as it is written in the manuscript, the sub-shelf basal melting rate perturbation allows for refreezing under your ice shelves, and in huge amount! In the Nordic seas you have a temperature difference interstadial minus stadial which is about +1 to +6°C. Your kappa is set to 5m/yr/°C in your standard OCN experiment which means that for negative beta you can easily end up with refreezing rate greater than 5 m/yr which seems completely off-scale (largely greater than snow accumulation). If refreezing is observed locally under ice shelves (due to recirculation of ice melt induced fresher waters along the ice), I think that a 40x40km shelf with more than 5 m/yr refreezing is completely unrealistic. I hope that I misinterpreted your equations, but if I am correct this is a serious flaw in your study. I may be wrong but I think that what we see in Fig 2d is due to your very large basal melting perturbation (with an amplitude st/is of more than 40 m/yr!): in the OCN experiment, your artificial refreezing allows for a rapid growth of ice shelves followed by a rapid disintegration. A side note: this is somehow rather peculiar to see that your ice sheet re-growth is actually faster than ice sheet collapse! Maybe you could show a figure showing the evolution of the spatially integrated value of the basal melt and how does this number compare with snow accumulation, ablation and calving rate. I strongly suggest the authors to use an oceanic perturbation written in a similar way to precipitation, based on a ratio of basal melting rate instead, preventing negative values.

- More generally, this is not clear to me if you distinguish correctly calving flux from melt. The two perturbation you applied (ATM and OCN) in the experiments impact the melt. Because you don't mention how your calving rate is calculated nor you give the extent of the ice shelves, it is difficult to quantify the respective role of melt vs. calving. Please provide the two fluxes separately in Fig. 2 and with the same units so that we

can clearly measure the impact of the oceanic perturbation on your calving flux. This addition would be very useful as basal melt cannot explain the IRD concentration in marine sediment cores. Also, as mentioned in my previous comment you should show the evolution in time of your basal melting rate along with the other components that explain the ice sheet volume evolution (surface ablation, accumulation and calving).

- I guess that in the model it does not matter if it comes from below or from above: melt is melt. It seems to me that if you get a larger response from the oceanic perturbation it is because your perturbation is also larger, am I right? I tried to get my answer from Fig. 4 but the color scale makes difficult to read the value of the basal melt along the coastlines: about 30 m/yr for the Scandinavian ice sheet and about 4 m/yr in the Bjørnøyrenna region (where you have no surface ablation at all)? If you impose a much larger oceanic perturbation than the atmospheric perturbation it is somehow expected to get a larger response? Please discuss. More or less related to this, how you maintain unconfined ice shelves with such high values of basal melting rates?

- Sub-shelf basal melting rate is not the only control exerted by the ocean on ice dynamics. What about sea level variability (and/or glacio-isostasy)? Some authors present the marine based Kara-Barents complex as an analogue for present-day West Antarctic ice sheet for which bedrock topography is a major control for stability. Of course marine ice sheet instability is generally triggered by a sub-shelf basal melt perturbation but is largely amplified by local bedrock depth with respect to sea level. In addition to provide more information on how your model deals with grounding line dynamics and glacio-isostasy, I think you should add a discussion about marine ice sheet instability of the Kara-Barents complex.

- Please provide more model information. SMB: what is the parameters used in the PDD model? Do you have a fixed daily variability (sigma)? Do you take into account refreezing? What is the value of your vertical lapse rate? GRISLI-UCM: how do you define the calving rate in the model? Maybe more importantly for ice sheet dynamics: how is computed the grounding line position? How do you combine SIA and SSA

approximations? You should also provide more information on your experimental setup: how is the ice sheet spun-up? Do you include glacio-isostasy? Do you have any kind of sea level forcing?

- You should assess the sensitivity of your results to the calving parametrisation / parameters.

- You justified the use of SST instead of sub-surface temperature because the Eurasian ice shelves are shallow. This is not really convincing. SST might be more correlated to surface processes (e.g. SMB) than to sub-shelf basal melting rates. Please provide a plot of sub-surface temperatures anomalies (Fig. 1) and a more robust justification for the use of the SST.

- Please improve on your figure quality. The plots are generally blurry (Fig. 3 and 4) and the color scales are not necessarily suited for the interpretation of the results (Fig. 4). The projection chosen is somehow unorthodox and you should draw the meridians and parallels.

Specific comments

- P1L14-16 please moderate: the larger response is expected as you impose a much larger oceanic perturbation compared to the atmospheric perturbation

- P3L24-26 these sentences are misleading: as you do not assess the impact of sea level variations and its impact on grounding line migration, you do not explicitly test "dynamic processes related to ice-ocean interactions". You quantify the effect of ice melt (and refreezing) scenarios on the dynamics of the ice sheet. Please rephrase.

- P4L8 What is the value of the proportionality factor?

- P4L9 How do you know where the sediment layer is saturated?

- P4L9-10 "explicitly calculates grounding line migration": how?

- P4L10 how calving is computed?

- P4L18 Please list the PDD model parameter values. Do you use an atmospheric lapse rate?

- P4L18-19 Please provide a reference or show the equation for the inland basal melting computation

- P4L19-20 A study from 2004 is not "recent"

- P6L6-8 I understand that the present day basal melting rate in the Arctic is difficult to quantify. However, you should present a map of B0 and B40k computed from your expression in order to quantify the role of kappa. This figure could also help to choose the right kappa value: being close to 0.1 at 40k and not too strong for present day (as we have sea-ice and you use SSTs).

- P6L25-26 You should show a map of ice thickness in the CTRL experiment clearly showing the extent of the grounded part of the ice sheet.

- P6L26 What is the depth of CLIMBER first oceanic level? Please justify better the use of SSTs.

- P7L4-7 Please show a map of the anomaly in surface ablation and in basal melt rate for beta=1 and beta=-1. This is important to quantify the imposed perturbation in your ATM and OCN experiment. Unlike Fig. 4, use for this the same topography (for example your spun-up initial topography).

- P7L13 Is this total ice volume? What is the volume of your spunup topography.

- P8L23-24 It is generally assumed that the melt anomaly is not linear with the temperature perturbation (e.g. Holland and Jenkins, J. of Climate, 2008). You should put more references in here and try to quantify your chosen sensitivity with respect to other melt models available in the literature. I agree that the basal melting rate is potentially highly variable but the fact that you use SST instead of sub-surface temperature added to the fact that you choose a high kappa value might lead to an overestimation of the oceanic induced melt sensitivity?

- P8L28-29 Apart from sub-shelf refreezing, what is the mechanism for ice-sheet re-growth? Please discuss in light of your ice volume gain of about 0.7 mSLE / 1000 years deduced from Fig 2d.

- P10L6 Be more specific: atmospheric and oceanic induced melt (you did not test the impact of sea level variations).

- P10L13-14 Show that this is still the case when you don't have refreezing under ice shelves.

- Fig 1 Annual means? Do you really have +12°C In Scandinavia at 40k?

- Fig 1 In this figure or in a new one: annual mean SMB anomaly (interstadial minus stadial) along with annual mean basal melting rate anomaly.

- Fig 2 Basal melting and surface ablation here as well, integrated over the whole ice sheet.

- Fig 2 Maybe in a separated plot: grounded and floating ice extent evolution for the different experiment

- Fig 2 2d Dashed lines?

- Fig 2 2d is this grounded or total ice volume ? Please show the floating ice.

- Fig 3 Which "stadials" and "interstadials" is represented here?

- Fig 3 Please clearly show the grounding line and the ice extent everywhere.

- Fig 3 Why the selected velocity point is not in the middle of the ice stream? Is it vertically integrated velocity?

- Fig 3 We see an increase in velocity in the ATM experiment but we cannot see anything in Fig 2. Why?

- Fig 3 Your ice shelf in stadials seem to have a very limited extent. Do you have certain specific boundary conditions, such as a depth criteria? If yes, this can be an

other problem as you will only have ice shelves if you start to retreat inland (which seems to be the case looking at your OCN (Is) snapshot). Conceptually, do you expect a larger ice shelf extent in interstadials relative to stadials?

- Fig 3 I am surprised to see that the British ice sheet presents generally very low velocities. I am guessing that it has a frozen bed, which is unexpected due to the warm climate in this area. Can you comment on this?

- Fig 3 In a separated plot: please show a map of ice thickness (with limit of grounded part) for the same selected glacial and respective interglacial as in Fig 2.

- Fig 4b I do not understand why there is a band of high basal melting rate (near the coastlines?). Also, why there is a wide area in the Nordic Seas with a relatively high basal melting rate: this area is not supposed to be grounded? Perhaps you have two different topographies here? This has to be clarified but I strongly suggest to plot the anomalies computed on the same ice sheet geometry (ideally the spun-up one). Please change the color scale.

- Fig 5 "region of Bjørnøyrenna": be more specific (maybe show this region on a map).

Technical corrections

- P7L26 SATs

- P9L22 amount

---

## Editor Comment (EC1) · N. Combourieu Nebout (Editor) · 5 Feb 2018

Dear authors,

We have now received the comment of the reviewers. Both reviews underline the potential of your research and the interest of your data. nevertheless one of them seems very critical about your tests. Take carefully attention on this point.

Please post your replies to all the comments on the discussion forum that explain how you want to amend your manuscript following the reviewer suggestions and answer to the critical comments. I would like to see your responses in track mode change.

[Figure]

I will give my decision on the basis of these documents.

Looking forward to reading your responses.

With my best regards

Nathalie Combourieu-Nebout

---

## Author Comment (AC1) · 25 Apr 2018

The study of Alvarez-Solas et al. is well written and of high scientific quality. Using an ice sheet model forced by idealised ocean and atmosphere climates through the last glacial period the authors investigate the response of the European Ice Sheet to millennial scale climate changes. A major finding of the study is that the European Ice Sheet, and in particular fast flowing ice in the vicinity of Bjørnøya trough in the Barents Sea, is highly sensitive to changes in ocean conditions with a minor contribution from changes in atmospheric surface mass balance.

The paper clearly merits publication in climate of the past as it provides important insight into the dynamics of Dansgaard-Oeschger events and the interplay of the ocean with marine terminating ice sheets of Eurasia. As opposed to previous studies focusing on the dynamics of the Laurentide Ice Sheet this work is novel in providing clues to the contrasting role of the Eurasian ice sheet to these millennial scale climate events of the last glacial. This said, there are a few important comments which have to be addressed before the manuscript can be accepted.

GENERAL COMMENTS:

1) The main conclusion of the work is that the response of the Eurasian Ice Sheet (EIS) during the last glacial period is dominated by changes to the ocean forcing in the Nordic Seas and Arctic Ocean. The study includes a sensitivity study testing the tuning factor used in controlling the ocean melting of ice. However, there is no assessment of the atmospheric forcing of the ice sheet as given by the model. In particular, what is the potential impact of different SMB parameterisations, different atmospheric climate realisations, and the potential impact of ablation on sub-glacial and basal and submarine melt (e.g. Bondzio et al., GRL, 2017).

We agree with the reviewer and thus have now studied the sensitivity of our results to the uncertainties in the atmospheric forcing by assessing the potential impact of different PDD parameters, the mentioned potential interplay between ablation and the sub-glacial state, and the response of the ice sheet to different oceanic sensitivities in terms of its dynamics (see below).

In relation with the assessment of the atmospheric forcing, the use of different atmospheric realisations is subject to the availability of climate simulations with different models for the three climate states needed: glacial (stadial), present, and interstadial. The latter is only available for a reduced number of models. For this reason we have not assessed this issue in the present study. Nevertheless, we think that an intercomparison among these should be in the scope of future work, and have mentioned it in the Discussion section: :

*"The use of different atmospheric realisations is subject to the availability of climate simulations with different models for the three climate states needed: glacial (stadial), present, and interstadial. The latter is only available for a reduced number of models. This makes the assessment of this issue difficult in the present study."*

Regarding the relationship between ablation and the ice dynamics, according to Bondzio et al, 2017, the displacement of the calving front is expected to cause an acceleration of ice streams (Jacobshavn in their case) both due to the direct impacts of a reduced back force and to the decrease of the viscosity near the shear margins.  The first of those phenomena is well captured by GRISLI-UCM. The second one is only indirectly captured (through the reduced viscosity from warming the ice due to an increase in the strain heating caused by the acceleration). Capturing it fully would require a complete consideration of the membrane stresses which is nowadays computationally not-affordable for long simulations and only feasible with full-Stokes or ice-flow models on local problems.

Another aspect discussed in Bondzio et al. (2017) concerns the effects that an increase in the microscopic water content would have on the viscosity furtherly increasing the acceleration of the ice stream through a thermomechanical feedback.   A direct computation of this phenomenon requires a sophisticated treatment of the non-linear rheology which should be in the scope of future work, in the frame of the development of a new ice-sheet model which we are carrying out in parallel (Robinson et al, in preparation).

Finally, to assess the uncertainty associated to the PDD approach (see below), the potential impact of ablation on the basal state of the ice sheet and thus on its dynamics, we performed a new ensemble of simulations considering different values of the refreezing parameter (csi) and the sensitivity to the ocean (kappa). Note that the proportion of the water produced from surface ablation that is allowed to refreeze in the ice does not directly change the viscosity in our model but the surface mass balance. Thus our ensemble is intended to illustrate the potential relationship between a larger ablation (for a given surface warming) and the dynamic response of the ice sheet to an oceanic perturbation. The analysis of this new ensemble shows, however, a negligible amplification effect from reduced (or increased) refreezing from surface waters on the dynamic behavior of the ice sheet.

This is now illustrated in figure S3. And the associated discussion has been treated in the supplementary text, and reads:

*"The potential interplay between the amount of refreezing from surface ablation and the response to an oceanic warming has also been investigated by considering different values of the refreezing parameter, $r$, and the sensitivity to the ocean, $\kappa$, (Figure S3).*

*Note that the proportion of the water produced from surface ablation that is allowed to refreeze in the ice does not directly change the viscosity in our model but the surface mass balance. Thus our ensemble is intended to illustrate the potential relationship between a larger ablation (for a given surface warming) and the dynamic response of the ice sheet to an oceanic*

*perturbation. The analysis of this new ensemble shows a negligible amplification effect from reduced (or increased) refreezing from surface waters on the dynamic behavior of the ice sheet."*

2) As stated in the manuscript, the authors use the PDD method to calculate surface ablation. However, the validity of the associated tuning parameters of this parameterisation for the LGM and stadial/interstadial climates investigated is not assessed or discussed. This analysis must be included in order to properly assess the relative role of atmospheric and oceanic forcing.

We understand the reviewer's concern with the validity of the PDD tuning parameters. Thus, we have studied the effects of varying the main parameters of the PDD model. These (together with the refreezing parameter, see above), and the explored values range are
$\sigma$: 4,5,6 K; f_PDD_snow: 0.0015, 0.003, 0.006 mwe/PDD; f_PDD_ice: 0.004, 0.008, 0.016 mwe/PDD

This new ensemble is illustrated in figure S2. This new figure shows the millennial-scale response to different PDD realisations. The amplitude of the FIS response (characterized through the standard deviation of the time series) varies from ~$2x10^{14}$ $m^3$ (for low values of f_PDD_ice, f_PDD_snow and $\sigma$ ) to ~$6x10^{14}$ $m^3$ (for high values of the parameters. Simulations in which the oceanic component has been activated show larger amplitudes for kappa > 3 m/yr/K, thus confirming the more important role the oceanic forcing plays on millennial scales compared to the atmosphere. The discussion of this aspect has now been addressed in the supplementary text, and reads:

*"To assess the uncertainty associated to the PDD approach, the potential impact of ablation on the basal state of the ice sheet and thus on its dynamics, we performed a new ensemble of simulations"*

*"Figure S2,right shows the variability of the 91 simulations exploring the uncertainty of the PDD model in terms of the standard deviation of the time series for the period 100 - 10 ky BP. This amplitude is compared with the one shown by exploring the values of $\kappa$ (from 1 to 9 m/yr/K) in a OCNsrf ensemble. A greater amplitude when forcing with the ocean is found from $\kappa$ = 3 /m/yr/K."*

Note that the potential impact of exploring the atmospheric uncertainties in our conclusions has now been deeply assessed. So, first we slightly modulated the strength of our conclusions regarding the major role the ocean plays with respect to the atmosphere. Second, the main conclusions being still robust, the analysis concerning the uncertainty of the parameterisations (both oceanic and atmospheric) is now shown in the Supplementary Information part of the paper.

3) The climate forcing used is not clearly assessed either. E.g. how do the MIS3 and stadial-interstadial climate changes from the model applied (figure 3) correspond to Observations?

This issue was partially already raised in the first General Comment. As stated above, we have not carried out an analysis of the uncertainty with respect to the climate forcing. However, in a different manuscript currently under review (Banderas et al. 2017) we have assessed the performance of the method that we use to built the synthetic climate forcing for the ice-sheet model, based on the use of the three climate states: glacial, interglacial and interstadial, as compared with previous approaches. The climatic forcing time series derived from these methods (temperature, precipitation) were compared at several locations with the available proxy data: the Greenland ice-core record and a reconstruction of temperature and precipitation based on delta18O variations from speleothems located in central Europe and southwestern North America, respectively. By construction, our method provides a perfect agreement with the ice-core record, improving the performance of previous methods. For temperature in Greenland the old and new methods follow a similar evolution, as dictated by the Greenland ice-core record, but elsewhere the new method shows a larger orbital and smaller millennial-scale amplitude. For precipitation, our new method yields a very different time evolution as a result of the spatial millennial-scale anomaly pattern which successfully reproduces the phasing and timing of delta18O variability in southwestern North America on millennial time scales, a result that cannot be achieved by the old method. Finally, in terms of the extent of NH ice sheets at the LGM, our new method appears to perform best, showing the most satisfactory agreement with reconstructions: ICE-5G (Peltier, 2004) for the LIS and DATED-1 (Hughes et al. 2016) for the FIS (Figure 4c; see also the Supplementary Material of Banderas et al. 2017). Thus we think that our method provides a clear improvement with respect to previous ones.

4) Also, the authors chose to apply SSTs as ocean forcing, whereas the ice shelves off the EIS reach a considerable depth (authors state 500m) where ocean temperatures will be significantly different from the surface. The authors should therefore assess the impact of using sub-surface ocean temperatures instead of SSTs for their results. Note that in previous studies by the authors (e.g. Alvarez-Solas et al., CP, 2011), the sub-surface ocean temperature is used as a forcing. Although it is postulated in the manuscript that sub-surface temperature is more correct for Laurentide type ice shelves, this has to be tested and thoroughly discussed.

We have now repeated the OCN and ALL experiments with a sub-surface forcing centered around 400-meters of depth. This new 2D-field is now included in the new figure 2 which now contains the surface oceanic field (called OCNsrf) and the subsurface oceanic field (OCNsub). According to this, panel d) of current Figure 2 shows the stadial-interstadial anomalies of the subsurface ocean temperatures. This new field has now been used to force the model again. New results are now included in all current figures.

The abstract has been accordingly modified, and now reads:

*"Our results show that the EIS responds with enhanced ice discharges in phase with interstadial warming in the North Atlantic when forced with the surface of the ocean. Conversely, the discharges are found to happen both during stadials and at the beginning of the interstadials when the subsurface of the ocean is considered."*

And:

*"While the atmospheric forcing alone is only able to produce modest iceberg discharges, warming of the ocean leads to higher rates of iceberg discharges as a result of relatively strong basal melting within the margins of the ice sheet."*

We have also changed accordingly the experimental setup section

*"In our simulations, a large part of the NE sector of the EIS is is marine based with shallow bedrock depths between 500~m and less than 100~m in several locations further south. It is therefore unknown whether this marine ice sheet was more susceptible to changes in the surface or the subsurface of the ocean. To investigate the effect of this uncertainty we decided to perform two different simulations considering different depths: one corresponding to the surface (OCNsrf) and the other one considering deeper (subsurface) oceanic waters by averaging temperatures within the range of 550 - 1050 m depth  (OCNsub). Therefore we hereafter distinguish between Delta T_mil^ocn for surface or subsurface millennial-scale temperature anomalies, respectively (Figure 2). The realism and convenience of applying one or the other is addressed in section 5.*"

And several places in the results section where we described the results of applying this new forcing (please see manuscript)

5) In the discussion the authors briefly comment on the difference between freshwater and IRD. This needs further detail. It would be an advantage to include a plot of the amount of meltwater released so as to compare it with the calving. I.e. how much discharge is calving versus melting and what are the different timescales of the two components of the mass balance? This is important in assessing the potential impact on ocean circulation and sea ice.

We have now rearranged Figures 1, 2 and 3 and included new figures (Figures 4, 5, 7 and 8 of the current manuscript). Figure 5 now focuses explicitly on the time series of the different mass balance terms: released meltwater, surface mass balance, calving and basal melting.
The manuscript includes now a description of this aspect, and reads:

*"The response of the EIS has been analyzed in terms of its mass balance decomposition for the all-forcing runs (Figure 5). The surface ocean temperature varies in phase with the atmosphere. Thus, during stadial-to-interstadial transitions the high negative values of dV/dt can be explained by the conjunction of an initial sharp increase in ablation together with pronounced increases in*

*basal melting and calving, which allow a large grounding line retreat in the Bjornoyrenna basin (Figure 5, mid panel).*

*The rate of ice loss by basal melting is similar to that resulting from the increase in ablation (as reflected in the surface mass balance) during the peak of a stadial-to-interstadial period. However, basal melting is much more efficient than surface mass balance in decreasing volume along the whole duration of an interstadial. This is due to the fact that ablation is restricted to the southern borders of the EIS. Thus, when the ice sheet has retreated to areas of no ablation, in spite of a slight further loss provided by the elevation feedback it rapidly equilibrates and a negative surface mass balance can not propagate further inland. In contrast, when enhanced basal melting from higher oceanic temperatures is applied, the associated retreat can propagate further inland occupying a large proportion of the Bjornoyrenna basin and facilitating high rates of volume loss (although similar in amplitude with respect to SMB) during the whole interstadial period (see the animation in the Supplementary Information)."*

SPECIFIC COMMENTS:

1) Page 2, Line 34-35: stated here that IRD peaks found at end of stadials (with reference to von Kreveld et al., 2000) - OBS! This requires an extremely well dated chronology or tephra to assess the phasing between ocean and ice cores - i.e. such inferences relating to timing are extremely important and must be documented carefully.

We agree and have thus refrased the statement concerning the conclusions of Kreveld et al. 200). It is now stated with more caution, and reads:

*"Correlating IRD occurrence with temperature changes registered in Greenland remains, however, difficult because it requires an extremely well dated chronology to assess the phasing between ocean sediments and ice cores."*

2) Page 5, Line 1-5: The forcing is composed of a difference between glacial and present day temperature (and Precip etc) with reference to CLIMBER model runs. However, in figures there is a reference to MIS3 (e.g. fig.1). The use of LGM vs MIS3, stadial and interstadial climate states should be better defined. E.e. what is the stadialmode? The same as LGM from CLIMBER? Also, the interstadial model (with intensified NADW) must be documented - why is NADW stronger and what are key differences with LGM/stadial?

Figure 1 refers to MIS3 (corresponding to 40 kya BP), because this is the time period representing our background climate conditions. As stated in the manuscript this initial state is to a certain extent arbitrary and only intended to provide stable initial conditions. We then superpose millennial-scale variations. For these, we indeed consider the stadial (interstadial) state to be the weak (strong) Atlantic Meridional Overturning circulation (AMOC) state obtained with the  CLIMBER3-alpha model under LGM boundary conditions.  In our study the stadial state is represented by a climate simulation of the LGM with CLIMBER-3α (Montoya and

Levermann, 2008) while the interstadial mode was taken from a recent glacial transient simulation performed with the same model under glacial climatic conditions, but with intensified NADW formation (Banderas et al., 2015). This state was achieved by forcing the model with enhanced atmospheric CO2 concentration levels, which in our model is found to lead to a northward shift of deep water formation and and intensification of the AMOC.As suggested by the referee, we have expanded the definition and explanation of our climate fields intended to represent the interstadial and stadial modes, and added the following to the manuscript:

"*The key differences between these climate modes are that in the stadial, North Atlantic Deep Water (NADW) formation is relatively weak and takes place south of Iceland. Accordingly the sea-ice front in the North Atlantic reaches 40S. In the interstadial state there is a northward shift and intensification of NADW formation. Northward oceanic heat transport increases, and the North Atlantic and surrounding areas warm at the interstadial relative to the stadial state, in particular the Nordic Seas. The simulated interstadial state is thus characterised by a more vigorous AMOC, deeper convective areas together with reduced sea ice in the Nordic Seas, and a temperature increase of up to 10 K in the North Atlantic relative to the stadial state, with a maximum anomaly in the Nordic Seas (Figure 2)*"

In order to clarify this aspect we have now split the old Figure 1 in two figures. New Figure 1 includes the climatic fields used to build the background equilibrium climate state used to produce the ice sheet spinup. We also included the initial ice-sheet state in a third panel of this figure. Current Figure 2 focuses on the fields used to perturb the equilibrium state and shows anomalies in temperature and precipitation induced by the atmosphere the surface of the ocean and the subsurface.

3) Page 5, Line 27-30: The spin-up of the ice sheet model needs to be clearly described. This is essential for the response of the model to changes in climate forcing. What choices were made for basal friction and ice temperature, rheology etc. What was the spin-up procedure used and how does this impact the results and response of the ice sheet? All these aspects are important to document.

Basal friction choices followed Alvarez-Solas et al, 2011, CP. A description of the rheology in the model can be found in Ritz et al, 2001. This information along with the references has now been added in the model section.

The spin-up procedure is now more clearly described in the experimental set-up section.
We have added the following to the manuscript in the Experimental Setup section:

"*The ice sheet was forced with the resulting climatologies for 100 kyr previous to the starting of the perturbations described below. This allows the vertical temperature profile within the ice sheet to be equilibrated with the climate*"

We also included a panel of the initial ice-sheet sate after the spin-up in current Figure 1.

4) Figure 1: What is shown here? Annual mean values? Clearly define MIS3 as well as the stadial-interstadial experiments (see notes above).

The referee is right. The spatial components of the forcing shown in Figure 1 have been calculated at annual resolution. This has been now specified in the figure caption. The timing of MIS 3 and the stadial interstadial experiments have been properly described in the new version of the manuscript (see point 2 above).

5) Figure 2: The Greenland ice core index (beta) is key to all the model simulations, but is not well described in the manuscript. The index is quite different from published NGRIP d18O and temperature conversions - it should be clearly stated why the authors choose not to use the normalised ice core temperature data? Also, why is the present value of the index nearly the same as that of the stadial/glacial value?

The index shows only variations on millennial time-scales, since variability whose periodicity is greater than 19 kyr (the beginning of the orbital spectrum) has been removed. The reason for this is that our study attempts to asses to effect of millennial-scale variability alone during the last glacial period on the Eurasian Ice Sheet (EIS). As explained above, our experimental setup consists of an initial (i.e., control-run) MIS3 simulation corresponding to 40 kya BP achieved by choosing a value of -0.1 for our orbital index (alpha-star). To investigate the effect of millennial scale variability we then impose millennial-scale variations as represented by the beta-star index. Only by separating orbital (given by alpha-star) from millennial-scale variations (given by beta-star) variations are we able to isolate the effect of the latter in the EIS, which is our goal. The present and glacial values of the beta-star index are similar simply because at those two stages the contribution of millennial-scale is similar (and small). Finally, a transient study considering together the orbital and millennial-scale forcing would also have been possible, but this is in the scope of future work. Note that this is explained in the manuscript in the Experimental Setup section, where we state:

"Our forcing method allows to investigate the response of the EIS solely to millennial-scale climate variability at MIS 3 by keeping constant the orbital component of the forcing ($\alpha^{\star} = \alpha^{\star}_{40K}$) and letting $\beta^{\star}$ vary throughout the LGP"

6) Figure 2: the ATM experiments shown hardly any oscillations. However, previous work by the the authors as well as other studies show binge-purge like oscillations of the ice sheets given a constant forcing. Why is there no self-sustained oscillations in the version of GRISLI applied here?

As explained by Alvarez-Solas et al. 2013 PNAS, in the hybrid (SIA +SSA version) of GRISLI (and GRISLI-UCM) there are no binge-purge-like oscillations. The inclusion of the longitudinal stresses together with the computation of the ice streams by the SSA uniquely (with no SIA

sliding) makes the ice-stream behavior much more stable than in SIA-alone models (see material and methods in Alvarez-Solas et al, 2013 PNAS). Nonetheless some internal variability is still present. This is now more clearly illustrated in the time series of Figure 3.

7) Figure 2: Define the dashed lines in the figure and give details in the caption.

Figure 2 is now improved (current Figure 3)

8) Figure 3: The two regions are named SW and NE. This should be clearly stated in caption of figure. However, if possible these names should be more descriptive - e.g something relating to Bjørnøyrenna would be more logical. Also define where this feature is on the map. What is yellow circle?

This has now been redone. We no longer focus on the SW region. Velocities shown in current figures 7, 8 and 9 correspond to the Bjørnøyrenna region of the Barents sea, that we have named Bjørnøyrenna basin and highlighted in current Figure 1.

9) Figure 3: Note that the land topo in figure 1 and 3 are different. Would be better to use one land topo - or comment on why different (ice sheet vs climate model) and potential impact of this on results.

Current Figures 1 and 6 show now the same topo.

10) Figure 5: Not necessary to repeat formula from main text in the caption.

We agree and have suppressed the formula here.

11) Figure 5: the discussion of the relationship between ice sheet height/velocity and grounding line retreat would benefit from including a discussion of changes in the position of the calving front through time. How does this differ from the grounding line and how does it relates to any of the assessed ice parameters?

The movements of the calving front usually are accompanied by a grounding line displacement. For some minor ice-shelf breakups this close relationship can be broken, but with almost no effects upstream inland. Thus we consider that the grounding line position is the best indicator for characterizing the dynamic behavior of the marine part of the EIS. We have included this in the text to make this clear in the Results section

"Note that changes in the position of the calving front are usually accompanied by a grounding line displacement. For some minor ice-shelf breakups this close relationship can be broken, but with almost no effects upstream inland. Thus we consider that the grounding line position is the best indicator for characterizing the dynamic behavior of the marine part of the EIS."

TECHNICAL COMMENTS: Line 21: Better to use DO-events and H-events or similar nomenclature. D/O is not a standard form.

Done. We have replaced this in the manuscript everywhere.

References:

Bondzio, J. H., Morlighem, M., Seroussi, H., Kleiner, T., Rückamp, M., Mouginot, J., Moon, T., Larour, E. Y., and Humbert, A.: The mechanisms behind Jakobshavn Isbræ's acceleration and mass loss: A 3-D thermomechanical model study, Geophysical Research Letters, 44, 6252–6260, https://doi.org/10.1002/2017GL073309, 2017GL073309, 2017.

Anonymous Referee #2

Alvarez-Solas et al. investigate the respective role of atmospheric forcing and oceanic induced sub-shelf basal melting (and refreezing?) rates in the variability of the Eurasian ice sheet during the last glacial period. The paper is scientifically very exciting as whilst a fair amount of climate records exist only little is known about glacial ice sheet variability and in particular the role of the ocean in this variability. However, I have serious doubt on the experimental setup and in particular concerning the basal melting perturbation chosen. If the authors really use an oceanic perturbation allowing for refreezing (and at a greater rate than snow accumulation!) the validity of the paper findings can be largely questioned. I suggest that the authors clarify their methodology as I will not support publication of this paper with the OCN experiment as presented.

I would recommend the authors to perform again their OCN and ALL experiment with
an alternative basal melting rate perturbation (e.g. based on a ratio as for precipitation
or at least with a positive threshold).

We opted for the simplest possible experimental setup, but we understand the reviewer's concerns. Thus we have followed the referee's suggestion and repeated all our main experiments (see below) limiting the potential accretion by one order of magnitude with respect to the melting, as recently suggested (Obase et al, 2017). As is shown our main conclusions still hold, in the sense that the ocean remain the major driver of ice-volume variations.

General comments

1) I am worry about your experimental setup: as it is written in the manuscript, the sub- shelf basal melting rate perturbation allows for refreezing under your ice shelves, and in huge

amount! In the Nordic seas you have a temperature difference interstadial minus stadial which is about +1 to +6 ∘ C. Your kappa is set to 5m/yr/ ∘ C in your standard OCN experiment which means that for negative beta you can easily end up with refreezing rate greater than 5 m/yr which seems completely off-scale (largely greater than snow accumulation). If refreezing is observed locally under ice shelves (due to recirculation of ice melt induced fresher waters along the ice), I think that a 40x40km shelf with more than 5 m/yr refreezing is completely unrealistic. I hope that I misinterpreted your equations, but if I am correct this is a serious flaw in your study. I may be wrong but I think that what we see in Fig 2d is due to your very large basal melting perturbation (with an amplitude st/is of more than 40 m/yr!): in the OCN experiment, your artificial refreezing allows for a rapid growth of ice shelves followed by a rapid disintegration. A side note: this is somehow rather peculiar to see that your ice sheet re-growth is actually faster than ice sheet collapse! Maybe you could show a figure showing the evolution of the spatially integrated value of the basal melt and how does this number compare with snow accumulation, ablation and calving rate. I strongly suggest the authors to use an oceanic perturbation written in a similar way to precipitation, based on a ratio of basal melting rate instead, preventing negative values.

The referee was right in pointing out that we were probably overestimating refreezing. We have accordingly changed the experimental design of the oceanic perturbation to avoid this. Following the recent paper of Obase et al. (2017), we have limited ice accretion by one order of magnitude compared to melting. The new approach is described now at the end of the experimental setup section:

"*Note that for $\kappa\beta\Delta T < -B0$ refreezing is allowed (B(t) < 0). Following Obase et al (2017), in order to avoid unrealistic values of ice accretion under the ice shelves, the sensitivity to the ocean is decreased by one order of magnitude when basal melting becomes negative…*"

By introducing this more realistic formulation, one can see that the peculiar behavior pointed out by the referee (a faster regrowth than collapse) has now disappeared in Figure 3. The new figure concerning the rate of volume changes for the different forcings (current Figure 4), clearly shows that during episodes of retreat its speed reaches more than 3 mm/yr, while during re-growth the rate is as much as 1 mm/yr. We have also added to the current manuscript a figure (new Figure 5) showing the evolution of the different terms of the mass balance: basal melt, surface mass balance and calving. Note that despite this change our main conclusions are not affected, in the sense that oceanic changes remain the major driver of ice volume changes. We have, nevertheless, modulated our conclusions in light of the referee's suggestion, and added the description of new Figure 4 showing the derivative of the volume in time:

"*The magnitude of these changes for the MIS 3 period is illustrated in Figure \ref{fig4}. The simulation forced with the surface of the ocean (OCNsrf) and that including the rest of the forcings (OCNsrf + ATM + SL) show the largest amplitudes, with peaks of sea-level rise above 4mm~yr$^{-1}$ during DO-events and sustained contributions well above 1~mm~yr$^{-1}$ during entire interstadial periods. In ATM, a decline of the EIS during stadial-to-interstadial*

*transitions is still observed but presents a smaller amplitude of 1-2~mm~yr$^{-1}$. The simulations in which the ice sheet is forced with the subsurface of the ocean present a decline of their volume during stadial periods and regrowth during interstadials as a consequence of the inverted spatial pattern of temperature anomalies with respect to the surface. In the case of OCNsub the amplitude of these changes is smaller than in the OCNsrf case, on the order of 0.5-1~mm~yr$^{-1}$, and reaches more than 1~mm~yr$^{-1}$ during pronounced stadials (as ca. at 44 ka BP). The OCNsub + ATM + SL simulation shows a slight volume loss during interstadials, as a consequence of the atmospheric forcing, that is superimposed onto the OCNsub behaviour."*

2) - More generally, this is not clear to me if you distinguish correctly calving flux from melt. The two perturbation you applied (ATM and OCN) in the experiments impact the melt. Because you don't mention how your calving rate is calculated nor you give the extent of the ice shelves, it is difficult to quantify the respective role of melt vs. calving. Please provide the two fluxes separately in Fig. 2 and with the same units so that we can clearly measure the impact of the oceanic perturbation on your calving flux. This addition would be very useful as basal melt cannot explain the IRD concentration in marine sediment cores. Also, as mentioned in my previous comment you should show the evolution in time of your basal melting rate along with the other components that explain the ice sheet volume evolution (surface ablation, accumulation and calving).

We have followed referee's suggestion and made a new figure containing all this information, with the two fluxes given separately in the same units. The new figure (Figure 5) shows the different components of the mass balance. It shows that, thanks to the new oceanic approach (see above), rates of oceanic-induced re-growth remain lower than those induced by the surface mass balance. The comparison of the contributions is discussed in the following point.

3) - I guess that in the model it does not matter if it comes from below or from above: melt is melt. It seems to me that if you get a larger response from the oceanic perturbation it is because your perturbation is also larger, am I right? I tried to get my answer from Fig. 4 but the color scale makes difficult to read the value of the basal melt along the coastlines: about 30 m/yr for the Scandinavian ice sheet and about 4 m/yr in the Bjørnøyrenna region (where you have no surface ablation at all)? If you impose a much larger oceanic perturbation than the atmospheric perturbation it is somehow expected to get a larger response? Please discuss. More or less related to this, how you maintain unconfined ice shelves with such high values of basal melting rates?

Concerning the caveat of "a larger perturbation, a larger response", accumulation and ablation compared to basal melting are now illustrated in Figure 5. The rate of ice loss by basal melting is similar to that resulting from the surface mass balance change (from an increase in ablation) during the peak of a stadial-to-interstadial period. However, basal melting is much more efficient than surface mass balance in decreasing volume along the whole duration of an interstadial (in

the case of OCNsrf) or a stadial (OCNsub). This is due to the fact that ablation is restricted to the southern borders of the EIS. Thus, when the ice sheet has retreated to areas of no ablation, in spite of a slight further loss provided by the elevation feedback, it rapidly equilibrates and a negative surface mass balance can not be propagated further inland. In contrast, when enhanced basal melting from higher oceanic temperatures is applied, the associated retreat can propagate further inland occupying a large proportion of the Bjonorema bassin and facilitating high rates of volume loss (although similar in amplitude with respect to SMB) during the whole interstadial (or stadial in the case of OCNsub). To make this clear this discussion has been added in the Results section:

*"The rate of ice loss by basal melting is similar to that resulting from the surface mass balance change one (from an increase in ablation) during the peak of a stadial-to-interstadial period. However, basal melting is much more efficient than surface mass balance in decreasing volume along the whole duration of an interstadial (in the case of OCNsrf) or a stadial (OCNsub). This is due to the fact that ablation is restricted to the southern borders of the EIS. Thus, when the ice sheet has retreated to areas of no ablation areas (in spite of a slight further loss provided by the elevation feedback) it rapidly equilibrates and a negative surface mass balance can not be propagated further inland. In contrast, when enhanced basal melting from higher oceanic temperatures is applied, the associated retreat can propagate further inland occupying a large proportion of the Bjonorema bassin and facilitating high rates of volume loss (although similar in amplitude with respect to SMB) during the whole interstadial (or stadial in the case of OCNsub)."*

Concerning ice shelves, the temporal extension of the ice shelves is now included in Figures 7 and 8. Their extension is greatly reduced during periods of enhanced basal melting as it is now illustrated in these figures. There are not large unconfined ice shelves surviving during these episodes. Some ice shelves can however survive during these episodes of high basal melting thanks to an increase in advection from the Bjonoram ice stream triggered by a grounding line retreat (see Figure 6). The grounding line has now more clearly been depicted so no misunderstanding between grounded and floating parts can happen. To make this clear we have made a new video that is now included in the Supplementary Information, and we have included this discussion in the Results section:

*"The extension of ice shelves is greatly reduced during periods of enhanced basal melting (Figures 7,8), with no large unconfined ice shelves surviving during these episodes (see also the Supplementary Information). Some ice shelves can however remain, in spite of the enhanced basal melting, thanks to an increase in advection from the Bjonoram ice stream triggered by a grounding line retreat (see Figure 6)."*

4) - Sub-shelf basal melting rate is not the only control exerted by the ocean on ice dynamics. What about sea level variability (and/or glacio-isostasy)? Some authors present the marine based Kara-Barents complex as an analogue for present-day West Antarctic ice sheet for which bedrock topography is a major control for stability. Of course marine ice sheet instability is generally triggered by a sub-shelf basal melt perturbation but is largely amplified by local

bedrock depth with respect to sea level. In addition to provide more information on how your model deals with grounding line dynamics and glacio-isostasy, I think you should add a discussion about marine ice sheet instability of the Kara-Barents complex.

We thank the referee for the suggestion of including a discussion on the Barents-Kara / Antarctica analogy. We have followed it and this discussion is now more explicitly addressed in the Discussion section.

*"Some authors present the marine based Kara-Barents complex as an analogue for present-day West Antarctic ice sheet for which bedrock topography is a major control for stability. Marine ice sheet instability is generally triggered by a sub-shelf basal melt perturbation but is largely amplified by local bedrock depth. We have shown, in this sense, that the Bjornoyrenna basin is highly susceptible to changes in the oceanic temperatures. The timing of this response with respect to changes registered in Greenland depends, however, on whether the surface or the subsurface of the ocean is considered as the relevant forcing of the ice sheet."*

We also explored the mentioned dependence on sea-level variability, and compare it to a control simulation (see new Figure 3). Changes in the imposed sea level appear not to be sufficient to cause any substantial change in EIS volume. The manuscript has been accordingly modified, including this aspect.

In the Offline forcing section we added:

*"Finally, millennial-scale sea-level variations are prescribed according to the reconstruction by Grant et al. (2012; Section 2.3). The specific details of the experimental setup used are described below".*

In the Experimental Setup section we added:

*In addition, varying sea-level forcing is considered (Figure 3b), both alone (SL run) and in combination with the previous forcings (ATM+OCN+SL).*

*"Under constant forcing, the CTRL run shows negligible millennial-scale sea-level equivalent (SLE) variations, although a lower frequency SLE fluctuation is found related to internal ice-sheet variability (Figure 3). When the model is forced only by changes in sea level, (SL run) a slight response is observed on millennial-scales. These changes appear not be sufficient to cause a substantial migration of the grounding line, thus not affecting ice velocities."*

5) - Please provide more model information. SMB: what is the parameters used in the PDD model? Do you have a fixed daily variability (sigma)? Do you take into account refreezing? What is the value of your vertical lapse rate?

The main parameters used in of the PDD model, for refreezing and the lapse rates are now described in the main text:

"*Its main parameters are the standard deviation of daily temperature, $\sigma$, and the conversion factors from PDDs to melt for snow and ice, f_PDD_snow and f_PDD_ ice. Here, sigma = 5 K, f_PDD_snow = 0.003 mwe/PDD and f_PDD_ice 0.008 mwe/PDD. Refreezing is considered, with a value of r = 60%. GRISLI-UCM accounts for changes in elevation at each time step considering a linear atmospheric vertical profile for temperature with different lapse rates in summer and in the annual mean (0.0080 and 0.0065 K/m, respectively) to account for the smaller summer atmospheric vertical stability (see also the Supplementary Information)".*

We have now also explored the sensitivity of our ice sheet model to the following range of values of these parameters:

**σ**: 4,5,6 K; f_PDD_snow: 0.0015, 0.003, 0.006 mwe/PDD; f_PDD_ice: 0.004, 0.008,0.016 mwe/PDD
This information has now been added in the Supplementary Information, where we address all the new sensitivity test devoted to explore the atmospheric uncertainties.

6) GRISLI-UCM: how do you define the calving rate in the model?
Maybe more importantly for ice sheet dynamics:
how is computed the grounding line position?

Calving is the result of a threshold criterion together with a sem-Lagrangian diagnosis of the advection on the ice shelf. The grounding line position is the result of applying the flotation criterium after the mass conservation equation is solved. This information has now been added to the new version of the manuscript in the Model description section, and reads:

"*The grounding line position dynamically evolves following the flotation criterion after the mass conservation equation is solved. Ice on the ice-shelf front calvs following a double criterion: Its thickness must first fall below a threshold (H_calv = 150 m , in the standard setup: see Supplementary Information for further information about the dependence of our results on this parameter), and second the upstream advection must fail to maintain the ice thickness above this threshold following a semi-Lagrangian approach (Peyaud et al, 2007)"*

7) How do you combine SIA and SSA?

SSA is systematically and only applied for regions of floating ice. Ice streams are simulated as "dragging" ice shelves. The criterium to activate SSA inland holds on the presence of water above 1 meter in places of soft sediments (Laske et al. 1997) and above 400 meters in absence

of these sediments. This information has now been included in the manuscript in the Model description , and reads:

*"The criterion to activate SSA inland holds on the presence of water above 1 meter in places of soft sediments (Laske et al. 1997) and above 400 meters in absence of these sediments"*

*And*

*"The effects of varying this proportionality factor on the simulated ice streams are discussed in Alvarez-Solas et al (2011)"*

Please see Alvarez-Solas et al, 2011 for further information.

8) - You should assess the sensitivity of your results to the calving parametrisation / parameters.

We have performed a new ensemble varying the main parameter controlling calving: the threshold in thickness below which the ice is calved. We have now explored a wide value range from 10 to 800 meters. Values of this threshold above 400 m allow an efficient disintegration of the Barents-Kara complex due to its relative shallow bed. The typical value of this parameter is however 150 meters (see Peyaud et al, 2007). The overall effect of this sensitivity test around the prefered value is to modulate the amplitude of the response to the oceanic perturbations (see figure below) but we think this does not change the main conclusions of this paper. A more thorough exploration of different calving laws is in the scope of future work that we are currently planning.

[Figure]

9) You justified the use of SST instead of sub-surface temperature because the Eurasian ice shelves are shallow. This is not really convincing. SST might be more correlated to surface processes (e.g. SMB) than to sub-shelf basal melting rates. Please provide a plot of sub-surface temperatures anomalies (Fig. 1) and a more robust justification for the use of the SST.

This issue was also raised by reviewer 1 and has motivated as pointed out above carrying out a new set of experiments. We have now repeated the OCN and ALL experiments with a sub-surface forcing centered around 400-meters of depth. This new 2D-field is now included in the new figure 2 which now contains the surface oceanic field (called OCNsrf) and the subsurface oceanic field (OCNsub). According to this, panel d) of current Figure 2 shows the stadial-interstadial anomalies of the subsurface ocean temperatures. This new field has now been used to force the model again. New results are now included in all current figures.

The abstract has been accordingly modified, and now reads:

*"Our results show that the EIS responds with enhanced ice discharges in phase with interstadial warming in the North Atlantic when forced with the surface of the ocean. Conversely, the discharges are found to happen both during stadials and at the beginning of the interstadials when the subsurface of the ocean is considered."*

And:

*"While the atmospheric forcing alone is only able to produce modest iceberg discharges, warming of the ocean leads to higher rates of iceberg discharges as a result of relatively strong basal melting within the margins of the ice sheet."*

We have also changed accordingly the Experimental setup section where we described the results of applying this new forcing.

*"To investigate the effect of this choice we decided to perform two different simulations considering different depths: one corresponding to the surface (OCNsrf) and the other one considering deeper (subsurface) oceanic waters by averaging temperatures within the range of 550 - 1050 m depth (OCNsub). Therefore we hereafter distinguish between Delta T_mil^ocn for surface or subsurface millennial-scale temperature anomalies, respectively (Figure 2). The realism and convenience of applying one or the other is addressed in section 5."*

Finally, we have included the corresponding results in the Results section:

*"In contrast, the oceanic forcing in OCNsrf induces pronounced changes in the dynamics of the EIS on millennial time scales, with episodes of a large volume reduction occurring during interstadials. The combination of sea level, atmospheric and oceanic forcings (SL + ATM +*

*OCNsrf run) results in a similar response of the EIS to that obtained in OCNsrf (Figure 3) as a consequence of the larger effect of the oceanic forcing in OCNsrf with respect to ATM. OCNsub shows an anti-phase relationship with respect to OCNsrf, with the largest reductions in ice volume occurring during prolonged stadial periods and regrowth phases happening during interstadials. This behavior can be explained by the fact that ocean waters at the subsurface warm (cool) during episodes of reduced (enhanced) convection at the Nordic Seas as a result of variations in the AMOC strength. When considering the forcing at the subsurface of the ocean together with the atmosphere (SL+ATM+OCNsub), slight reductions of the EIS volume (less than 1 m of s.l.e) during interstadials are superimposed to the previous behavior (Figure 3)"*

10) Please improve on your figure quality. The plots are generally blurry (Fig. 3 and 4) and the color scales are not necessarily suited for the interpretation of the results (Fig. 4). The projection chosen is somehow unorthodox and you should draw the meridians and parallels.

We have followed the reviewer's suggestion and modified all figures.

Specific comments

1) - P1L14-16 please moderate: the larger response is expected as you impose a much larger oceanic perturbation compared to the atmospheric perturbation

We have modulated this discussion now:

*"While the atmospheric forcing alone is only able to produce modest iceberg discharges, warming of the ocean leads to higher rates of iceberg discharges as a result of relatively strong basal melting within the margins of the ice sheet"*

And also moderated the Abstract and Conclusions in several places.

Nevertheless, we should point out here that we do not impose a larger oceanic perturbation per se. The temporal index is the same for the atmosphere and the ocean and the amplitude is given by an OGCM simulation of two different oceanic states mimicking a stadial and an interstadial. We then translate those fields into ablation (through PDD, whose uncertainty has now been largely explored, see General Comment 5 and response to Reviewer 1) and into basal melting (through a linear equation). The values of the oceanic sensitivity parameter (kappa) we used here are in the range (or even below in most cases) of those suggested by data in Antarctica (Rignot et al. 2002). Note, in particular, that even for low-mid values of kappa of 3 meter/year/Kelvin the response to the ocean appears to be of greater amplitude than that to the atmosphere, making our main conclusions robust. We have added this discussion as well in the Discussion section to make this point clear:

*"Our results indicate that the ocean is the major driver of the EIS ice-volume changes during MIS-3. The larger response to the could be expected to result from imposing a much larger oceanic perturbation compared to the atmospheric perturbation. However, we note that the temporal index used is the same for the atmosphere and the ocean and the amplitude is given by a OGCM simulation of two different oceanic states mimicking a stadial and an interstadial. We then translate those fields into ablation (through PDD, whose uncertainty has now been largely explored) and into basal melting (through a linear equation). The values of the oceanic sensitivity parameter (kappa) we used here are in the range (or even below in most cases) of those suggested by data in Antarctica (Rignot and Jacobs 2002). Note, in particular, that even for low-mid values of kappa of 3 meter/year/Kelvin the response to the ocean appears to be of greater amplitude than that to the atmosphere, making our main conclusions robust"*

2) - P3L24-26 these sentences are misleading: as you do not assess the impact of sea level variations and its impact on grounding line migration, you do not explicitly test "dynamic processes related to ice-ocean interactions". You quantify the effect of ice melt (and refreezing) scenarios on the dynamics of the ice sheet. Please rephrase.

This has now been done (please see the responses to General Comment number 4 above).

3) - P4L8 What is the value of the proportionality factor?

Please see the response to General Comment 7. We refer here to Alvarez-Solas et al, 2011, CP where this dependence is described and tested.

We have added the following sentence to the manuscript:

*"The effects of varying this proportionality factor on the simulated ice streams are discussed in Alvarez-Solas et al (2011)"*

4) - P4L9 How do you know where the sediment layer is saturated?

The water content of the sediment layer is prognosed by the model. The sediment is considered to be saturated in the presence of more than 1 meter of water. Please see Alvarez-Solas et al, 2011 for further information. We have clarified this in the Model description section and added the reference to the sediment map on which this is based (Laske et al, 1997)

*"The criterium to activate SSA inland holds on the presence of water above 1 meter in places of soft sediments (Laske et al. 1997) and above 400 meters in absence of these sediments"*

5) - P4L9-10 "explicitly calculates grounding line migration": how?

Please see the response to General Comment 6 above.

6) - P4L10 how calving is computed?
Please see the response to  General Comment 6 above.

7) - P4L18 Please list the PDD model parameter values. Do you use an atmospheric lapse rate?
Please see the response to General Comment 5 above.

8) - P4L18-19 Please provide a reference or show the equation for the inland basal melting computation

This information can be found in Ritz et al., (2001). We have added the reference in the pertinent place.

9) - P4L19-20 A study from 2004 is not "recent"
Recent has been deleted.

10) - P6L6-8 I understand that the present day basal melting rate in the Arctic is difficult to quantify. However, you should present a map of B0 and B40k computed from your expression in order to quantify the role of kappa. This figure could also help to choose the right kappa value: being close to 0.1 at 40k and not too strong for present day (as we have sea-ice and you use SSTs).

The suggestion made her by the referee is indeed one possibility, but we consider than calibrating B40 with respect to B0 via comparison to the present-day distribution of sea ice is difficult and that our simpler approach is still pertinent for the purposes of our paper.

11) - P6L25-26 You should show a map of ice thickness in the CTRL experiment clearly showing the extent of the grounded part of the ice sheet.

We followed referee's suggestion and added the 2D CTRL map in current Figure 1

12) - P6L26 What is the depth of CLIMBER first oceanic level? Please justify better the use of SSTs.

The first oceanic level of CLIMBER-3alpha goes from 0-50 meters depth. Concerning SSTs versus subsurface, please see General Comment number 9 above.

13) - P7L4-7 Please show a map of the anomaly in surface ablation and in basal melt rate for beta=1 and beta=-1. This is important to quantify the imposed perturbation in your ATM and

OCN experiment. Unlike Fig. 4, use for this the same topography (for example your spun-up initial topography).

The anomalies of ablation and basal melt are subjected to feedbacks triggered by the thermomechanical state of the ice sheet. Therefore a quantification of the atmospheric and oceanic perturbations can not simply be made by plotting those fields for a given beta (see also the response to GC 3).

Nevertheless the required information for quantifying the perturbations can now been found in a new figure. Thus, this caveat is now addressed by adding a figure concerning the mass balance terms (see Figure 5).

14) - P7L13 Is this total ice volume? What is the volume of your spunup topography.

Yes, this is the total (grounded plus floating) EIS SLE change. The volume of our spunup topography is 8.3 x 10^15 m^3.

15) - P8L23-24 It is generally assumed that the melt anomaly is not linear with the temperature perturbation (e.g. Holland and Jenkins, J. of Climate, 2008). You should put more references in here and try to quantify your chosen sensitivity with respect to other melt models available in the literature. I agree that the basal melting rate is potentially highly variable but the fact that you use SST instead of sub-surface temperature added to the fact that you choose a high kappa value might lead to an overestimation of the oceanic induced melt sensitivity?

The discussion on the submarine melting has been extended, and our choice justified, as follows:

"Several marine basal melting rate parameterizations can be found in the literature. The submarine melt rate is thought to be directly influenced by the oceanic temperature variations below the ice shelves. Accordingly, most basal melting parameterizations are built as function of the difference between the oceanic temperature at the ice–ocean boundary layer and the temperature at the ice-shelf base, generally assumed to be at the freezing point. The dependence on this temperature difference can be linear (Beckmann and Goosse, 2003) or quadratic (Holland et al., 2008b; Pollard and DeConto, 2012; DeConto and Pollard, 2016; Pattyn, 2017). The linear marine basal melting rate parameterization used in this study is the simplest case that allows testing of the ice-sheet sensitivity to past oceanic temperature changes. Nevertheless, it accounts separately for sub-ice-shelf areas near the grounding line and for purely floating ice (ice shelves). the basal melting rate for purely floating ice shelves (Bsh) is given by the grounding-line basal melt Bgl scaled by a constant factor $\gamma$ : Bsh(t) = $\gamma$ Bgl(t). In this study, $\gamma$ is set to 0.1. Thus, we consider that the submarine melting rate for ice shelves is 10 times lower than that close to the grounding zone, which is qualitatively in agreement with observations in some Greenland glaciers (Münchow et al., 2014; Rignot and Steffen, 2008; Wilson et al., 2017)"

The SST aspect is now included by the new experiments showing the response to subsurface temperatures as well.

16) - P8L28-29 Apart from sub-shelf refreezing, what is the mechanism for ice-sheet re-growth? Please discuss in light of your ice volume gain of about 0.7 mSLE / 1000 years deduced from Fig 2d.

Please see new figures 4 and 5. We also have explicitly mentioned the mechanism for regrowth in the results section.

Concerning OCNsrf:

*"During stadial periods, both enhanced positive mass balance and negative oceanic anomalies (producing a slight refreezing) favor the regrowth of the EIS."*

And concerning OCNsub:

*"Subsequently, reduced basal melting in the NE part of the EIS (allowing even a slight refreezing at its grounding line) favors regrowth of the Bj{\o}rn{\o}yrenna basin during interstadial periods."*

17) - P10L6 Be more specific: atmospheric and oceanic induced melt (you did not test the impact of sea level variations).

The impact of sea-level variations has now been included; please see General Comment number 4 and Specific Point number 2 above.

18) - P10L13-14 Show that this is still the case when you don't have refreezing under ice Shelves.

We have modulated this sentence. The new sentence reads now:

*"[...]. Added to the smaller contribution of the SW retreat, this results in sea-level changes on the order of several meters."*

Nevertheless, as explained in the new manuscript, refreezing is now decreased by one order and magnitude and new Figures 4 and 5 show that a large refreezing is not a necessary condition for the mentioned sentence to be true.

19) - Fig 1 Annual means? Do you really have +12 ◦ C In Scandinavia at 40k?

This was a mistake in the previous plot. It has now been corrected.

20) - Fig 1 In this figure or in a new one: annual mean SMB anomaly (interstadial minus stadial) along with annual mean basal melting rate anomaly.

Please see the response to Specific Comment 13.

21) - Fig 2 Basal melting and surface ablation here as well, integrated over the whole ice Sheet.

Please see new figure 5 and see the response to Specific Comment 13.

22) - Fig 2 Maybe in a separated plot: grounded and floating ice extent evolution for the different experiment

Done in new Figures 7 and 8.

23) - Fig 2 2d Dashed lines?

This figure has been remade and improved

24) - Fig 2 2d is this grounded or total ice volume ? Please show the floating ice.

It was the total ice volume. Floating ice is shown now in new Figures 7 and 8.

25) - Fig 3 Which "stadials" and "interstadials" is represented here?

The stadial corresponds to 45.6 ka BO and the interstadial corresponds to DO 12 (ca. 47 ka BP) and the This has been now included in the Figure caption.

26) - Fig 3 Please clearly show the grounding line and the ice extent everywhere.

All the 2D spatial plots are now re-done and improved.

27) - Fig 3 Why the selected velocity point is not in the middle of the ice stream? Is it vertically integrated velocity?

This has now been changed. We consider velocities of the whole basin. And yes, it is vertically integrated velocity.

28) - Fig 3 We see an increase in velocity in the ATM experiment but we cannot see anything in Fig 2. Why?

This is was a consequence of considering a grid point which was not located right in the middle of the ice stream. However a small velocity increase of ~0.1 km a$^{-1}$ is observed at the approximate timing of the interstadial associated to DO 12. The associated figures have now been re-done.

29) - Fig 3 Your ice shelf in stadials seem to have a very limited extent. Do you have certain specific boundary conditions, such as a depth criteria? If yes, this can be another problem as you will only have ice shelves if you start to retreat inland (which seems to be the case looking at your OCN (Is) snapshot). Conceptually, do you expect a larger ice shelf extent in interstadials relative to stadials?

We only consider depth for basal melting as an additional boundary condition above 750 meters, a deeper (thus more conservative) value than in Peyaud et al. 2007. This has been now included in the Model description:

*"The melt rate in the open ocean, that is considered as being beyond the continental shelf break, is prescribed to a high value (20 m a−1 ) to avoid unrealistic ice growth beyond 750 m of ocean depth, following Peyaud et al. (2007)"*

30) - Fig 3 I am surprised to see that the British ice sheet presents generally very low velocities. I am guessing that it has a frozen bed, which is unexpected due to the warm climate in this area. Can you comment on this?

For some periods of ice expansion (either from reduced ablation or a decreased in basal melting) the initial low thickness of the ice sheet in this region does not allow to isolate enough the base from negative temperatures at the surface (the climate is warm but presents low but negative values of its temperature during stadials). Therefore the base can be frozen and velocities low. Nevertheless this is not the common situation. Velocities are generally high in this region (please see new Figure 6 and the supplementary movies).

31) - Fig 3 In a separated plot: please show a map of ice thickness (with limit of grounded part) for the same selected glacial and respective interglacial as in Fig 2.

We do not understand here the precise request. We have nevertheless now clearly illustrated the position of the grounding line in new Figure 6.

32)- Fig 4b I do not understand why there is a band of high basal melting rate (near the coastlines?). Also, why there is a wide area in the Nordic Seas with a relatively high basal melting rate: this area is not supposed to be grounded? Perhaps you have two different topographies here? This has to be clarified but I strongly suggest to plot the anomalies computed on the same ice sheet geometry (ideally the spun-up one). Please change the color scale.

The reason for this is that the basal melting rate for purely floating ice shelves (Bsh) is given by the grounding-line basal melt Bgl scaled by a constant factor $\gamma$ : Bsh(t) = $\gamma$ Bgl(t). In this study, $\gamma$ is set to 0.1. Thus, we consider that the submarine melting rate for ice shelves is 10 times lower than that close to the grounding zone, which is qualitatively in agreement with observations in some Greenland glaciers (Münchow et al., 2014; Rignot and Steffen, 2008; Wilson et al., 2017). This discussion has been now included in the Model description:

"*The marine basal melting rate parameterization used in this work follows a linear approach that accounts separately for sub-ice-shelf areas near the grounding line and for purely floating ice (ice shelves). the basal melting rate for purely floating ice shelves (Bsh) is given by the grounding-line basal melt Bgl scaled by a constant factor $\gamma$ : Bsh(t) = $\gamma$ Bgl(t). In this study, $\gamma$ is set to 0.1. Thus, we consider that the submarine melting rate for ice shelves is 10 times lower than that close to the grounding zone, which is qualitatively in agreement with observations in some Greenland glaciers (Münchow et al., 2014; Rignot and Steffen, 2008; Wilson et al., 2017)*"

33) - Fig 5 "region of Bjørnøyrenna": be more specific (maybe show this region on a map).
This is now shown in new Figure 1.

Technical corrections

1) - P7L26 SATs

Done

2) - P9L22 amount

Done

References:
Obase, T., Abe-Ouchi, A., Kusahara, K., Hasumi, H., & Ohgaito, R. (2017). Responses of Basal Melting of Antarctic Ice Shelves to the Climatic Forcing of the Last Glacial Maximum and CO2 Doubling. *Journal of Climate*, *30*(10), 3473-3497.

---

## Author Comment (AC2) · 25 Apr 2018

The comment was uploaded in the form of a supplement:
https://www.clim-past-discuss.net/cp-2017-143/cp-2017-143-AC2-supplement.pdf

---

## Author Comment (AC3) · 25 Apr 2018

**Oceanic forcing of the Eurasian Ice Sheet on millennial time scales during the Last Glacial Period**

Jorge Alvarez-Solas1,2, Rubén Banderas1,2, Alexander Robinson1,2,3,4, and Marisa Montoya1,2

1Dpto. Astrofísica y Ciencias de la Atmósfera; Facultad de Ciencias Físicas; Universidad Complutense de Madrid (UCM) 2Instituto de Geociencias (UCM-CSIC), Madrid, Spain

3Potsdam Institute for Climate Impact Research (PIK), Potsdam, Germany

4Faculty of Geology and Geoenvironment, National and Kapodistrian University of Athens, Greece

Correspondence to: Jorge Alvarez-Solas (jorge.alvarez.solas@fis.ucm.es)

**Abstract.**

The last glacial period (LGP; ca.110-10 ka BP) was marked by the existence of two types of abrupt climatic changes, Dansgaard-Oeschger (DO) and Heinrich (H) events. Although the mechanisms behind these are not fully understood, it is generally accepted that the presence of ice sheets played an important role in their occurrence. While an important effort has

- 5 been made to investigate the dynamics and evolution of the Laurentide Ice Sheet (LIS) during this period, the Eurasian Ice Sheet (EIS) has not received much attention, in particular from a modeling perspective. However, meltwater discharge from this and other ice sheets surrounding the Nordic Seas is often implied as a potential cause of ocean instabilities that lead to glacial abrupt climate changes. Thus, a better understanding of its variations during the LGP is important to understand its role in glacial abrupt climate changes. Here we investigate the response of the EIS to millennial-scale climate variability during the
- 10 LGP. We use a hybrid, three-dimensional, thermomechanical ice-sheet model that includes ice shelves and ice streams. The model is forced offline through a novel perturbative approach that includes the effect of both atmospheric and oceanic variations and provides a more realistic treatment of millennial-scale climatic variability than conventional methods. Rev. 1 General Comment (GC) 4: Our results show that the EIS responds with enhanced ice discharge in phase with interstadial warming in the North Atlantic when forced with surface ocean temperatures. Conversely, when subsurface ocean temperatures are used,
- 15 enhanced ice discharge occurs both during stadials and at the beginning of the interstadials. Separating the atmospheric and oceanic effects demonstrates the major role of the ocean in controlling the dynamics of the EIS on millennial time scales. Rev. 1 GC4 and Rev. 2 GC9: While the atmospheric forcing alone is only able to produce modest iceberg discharges, warming of the ocean leads to higher rates of iceberg discharges as a result of relatively strong basal melting at the margins of the ice sheet. Together with previous work, our results provide a consistent explanation for the response of the LIS and the EIS to glacial
- 20 abrupt climate changes, and highlight the need for stronger constraints on the local North Atlantic behavior in order to improve our understanding of ice sheet's glacial dynamics.

**1 Introduction**

The last glacial period (LGP; ca.110-10 ka before present, BP) was marked by the existence of two types of abrupt climatic changes: Dansgaard-Oeschger (DO) and Heinrich (H) events (e.g. Alley et al., 1999). DO-events are identified in Greenland ice-core records as regional abrupt warmings by up to 16°C (Huber et al., 2006; Kindler et al., 2014) from cold (stadial)

- 5 to relatively warm (interstadial) conditions within decades (Dansgaard et al., 1993) followed by a gradual cooling interval lasting from centuries to millennia and an ultimate phase of rapid cooling back to stadial conditions (Steffensen et al., 2008). Superimposed on the millennial-scale variability associated with DO-events, an additional lower-frequency climatic cycle is identified. So-called Bond cycles are flanked by prolonged stadials ending with prominent DO-events within about 7-10 kyr (Bond et al., 1993). Preceding these, and concomitant with the culmination of the prolonged stadials, H-events are registered
- 10 in North Atlantic marine sediments as layers of remarkably high concentrations of ice-rafted debris (IRD) (Heinrich, 1988) as a result of massive iceberg discharges from the Laurentide ice-sheet (LIS) (Hemming, 2004).

While significant effort has been invested in understanding the role of the LIS in glacial abrupt climate changes, the dynamics of the Eurasian Ice Sheet (EIS) during the LGP has received comparatively less attention from a modeling perspective. However, improving our understanding of its evolution and response to past climate changes is important for a number of

- 15 reasons. First, constraining freshwater inputs into the North Atlantic Ocean is crucial for a better understanding of the driving mechanisms of glacial abrupt climate changes (Rasmussen and Thomsen, 2013), since meltwater discharge from the ice sheets surrounding the Nordic Seas is often implied as a cause of ocean instabilities. Precursor events could possibly have originated from the European and Icelandic ice sheets (Grousset et al., 2000; Scourse et al., 2000). Meltwater peaks in the Norwegian Sea during Marine Isotopic Stage 3 (MIS 3) have been associated with H events and millennial-scale climate variability (Lekens
- et al., 2006). From a broader perspective, the EIS, consisting of the Fennoscandian, the British Isles and the Barents-Kara ice sheets (FIS, BIIS and BKSIS, respectively) contained a large marine-based sector at its maximum extension (Hughes et al., 2016) that was exposed to oceanic variations, and the BKSIS is often considered as an analog for the current West Antarctic ice sheet (WAIS). At the LGM both had a similar size, but while the WAIS endured the deglaciation, the BKSIS completely disappeared (Andreassen and Winsborrow, 2009). Understanding the underlying mechanisms would provide important insights
  into the future evolution of the WAIS (Gudlaugsson et al., 2013, 2017).

Reconstructing the EIS response to past glacial abrupt climate changes prior to the LGM has been difficult, in part because, in reaching its maximum extent, the ice sheet eroded and removed nearly all older deposits. Nevertheless, the available paleodata indicate that during MIS 3 the EIS was highly dynamic, with its advance and retreat closely linked to stadials and interstadials. In this line, records from Norway (Mangerud et al., 2003, 2010; Olsen et al., 2002), Finland (Helmens and Engels, 2010) and

30 Sweden (Wohlfarth, 2010) indicate rapid and rythmic ice-sheet variations in western Scandinavia, with advances and retreats during stadials and interstadials, respectively. Recent records also indicate enhanced meltwater discharges during interstadials from the Svalbard-Barents Sea ice sheet and probably also from the Scandinavian ice sheet (Rasmussen and Thomsen, 2013). The resolution and quality of geophysical data across marine sectors has improved considerably in the past decade (Hughes et al. (2016) and references therein). The results confirm substantial variations of the EIS volume, with the largest uncertainties

in marine sectors of the ice sheets. Strong variations in the deposition of IRD suggests high co-variability of the BIIS with changes in ocean sea surface temperature (Hall et al., 2011; Scourse et al., 2009) and variations in EIS ice streams (Becker et al., 2017). North Atlantic marine sediment records register widespread variations of IRD input throughout the LGP indicating variations of iceberg rafting from virtually all surrounding ice sheets. Sources and timing differ among different sites. A

- 5 dominant periodicity equal to that of DO-events was identified in the Irminger Sea, with the largest IRD peaks at the end of stadials originating in the Iceland and Greenland ice sheets (von Kreveld et al., 2000). Strong millennial-scale iceberg rafting variability of the BIIS has been documented as well in the North Sea (Hall et al., 2011; Peck et al., 2007; Scourse et al., 2009), but enhanced IRD seems to occur both during interstadials and stadials. For the FIS, IRD records in the Norwegian Sea show the characteristic DO periodicity, with IRD discharge occurring just before interstadial transitions (Lekens et al., 2006).
- 10 More recently, however, an increase in IRDs from Fennoscandia during interstadials has been reported (Dokken et al., 2013; Becker et al., 2017). Rev. 1 Specific comment (SC) 1: Correlating IRD occurrence with temperature changes registered in Greenland remains difficult, however, because it requires an extremely well dated chronology to assess the phasing between ocean sediments and ice cores.
- Progress has been achieved also in the past decade using ice-sheet models. Siegert and Dowdeswell (2004) used inverse modelling to simulate the EIS evolution during the second part of the LGP, matching the geological evidence presented by optimizing the fit with data. Forsström and Greve (2004) used subsequent versions of a three-dimensional, polythermal icesheet model to simulate the EIS evolution throughout the LGP. Important variations in the EIS ice volume in response to temperature and precipitation variations were simulated. Clason et al. (2014) additionally included a parameterisation of surface meltwater enhanced sliding. In both cases too much ice was simulated in the northeastern EIS. Gudlaugsson et al. (2017)
- 20 used the same model but introducing a simple representation of the subglacial hydrological system, focusing on its role in the temporal evolution of the EIS. Recently, an ice-sheet model constrained by data has been used to simulate the EIS evolution throughout part of LGP (Patton et al., 2016). The model targets the most probable EIS distribution at different time slices and reproduces substantial ice-volume variations. However, all of these models suffer from limitations, such as the use of the shallow-ice approximation (SIA) and its associated lack of an explicit treatment of the oceanic forcing. Marshall and Koutnik
- 25 (2006) investigated the production of icebergs from all the North American ice sheets with a parameterized calving model. They found different behaviors on millennial time-scales depending on the local glaciological and climatic characteristic, with increased iceberg production both during stadials (e.g. from Iceland) or during interstadials (e.g. from Barents Sea). Nonetheless, sub-marine melting at the grounding line has not been explicitly considered until now and its impacts on millennial-scale variability have not been investigated up to now from a modelling perspective.
- 30 Here, we investigate the response of the EIS to millennial-scale climate variability during MIS 3 using a three-dimensional ice-sheet model. To this end, a novel offline approach is used that provides a better representation of millennial-scale climate variability (Banderas et al., 2017). In addition, for the first time, both the atmospheric and oceanic effects of millennial scale climate variability associated with glacial abrupt climate changes are considered. This facilitates the quantification of the relative contribution of surface (ablation) and dynamic processes related to ice-ocean interactions.

The paper is organized as follows: in Section 2 the ice-sheet model, the forcing method and the experimental setup are described. In Section 3 the response of the EIS to the imposed forcing is shown, the focus being the evolution of its ice volume, its impact on sea level and the mechanisms behind meltwater and ice discharge. Finally, the main conclusions are summarised in Section 5.

**5 2 Model and experimental setup**

**2.1 Model**

10

The model used in this study is the ice-sheet model GRISLI-UCM, an extension of the original model GRISLI developed by Ritz et al. (2001). GRISLI-UCM is a hybrid three-dimensional thermomechanical ice-sheet model. Inland ice flows through deformation under the Shallow Ice Approximation (SIA, Hutter, 1983). Ice shelves and ice streams are described following the Shallow Shelf Approximation (SSA, MacAyeal, 1989). Ice streams (areas of fast flow, typically faster than  $10^2 m a^{-1}$ ) are considered as dragging ice shelves, allowing for basal movement of the ice (Bueler and Brown, 2009). Basal stress under ice streams is proportional to ice velocity and to the effective pressure of ice. **Rev. 2 GC7 and SC3:** The effects of varying

- this proportionality factor on the simulated ice streams are discussed in Alvarez-Solas et al. (2011). The locations of the ice streams are determined by the presence of basal water within areas where the sediment layer is saturated. Rev. 2 GC7
  and SC4: The criterion to activate SSA inland relies on the presence of water above 1 meter in places of soft sediments (Laske, 1997) and above 400 meters in absence of such sediments. Rev. 2 GC6, SC5 and SC6: The grounding line position dynamically evolves following the flotation criterion after the mass conservation equation is solved. Ice on the ice-shelf front calves following a two criteria. First, its thickness must first fall below a threshold (*Hcalv* = 150 m, in the standard setup; see
- 20 the upstream advection must fail to maintain the ice thickness above this threshold following a semi-Lagrangian approach (Peyaud et al., 2007). GRISLI-UCM thus explicitly calculates grounding line migration, ice-stream and ice-shelf velocities. This allows the model to properly represent both grounded and floating ice. GRISLI-UCM uses finite differences on a staggered Cartesian grid at a 40 km resolution, corresponding to 224×208 grid points for the Northern Hemisphere domain, including the EIS, with 21 vertical levels. By default, initial topographic conditions are provided by surface and bedrock elevations built

the Supplementary Information for further information regarding the dependence of our results on this parameter). Second,

- 25 from the ETOPO1 dataset (Amante and Eakins, 2009) and ice thickness (Bamber et al., 2001). The surface mass balance is given by the sum of accumulation and ablation, both of which are calculated from monthly surface air temperatures (SATs) and monthly total precipitation. Accumulation is calculated by assuming that the fraction of solid precipitation is proportional to the fraction of the year with mean daily temperature below 2°C. The daily temperature is computed from monthly SATs assuming that the annual temperature cycle follows a cosine function. Ablation is calculated using the positive-degree-day (PDD) method
- 30 (Reeh, 1989). Rev. 2 GC5 and SC7 : Its main parameters are the standard deviation of daily temperature,  $\sigma$ , and the conversion factors from PDDs to melt for snow and ice,  $f_{PDD_{snow}}$  and  $f_{PDD_{ice}}$ . Here,  $\sigma = 5$  K,  $f_{PDD_{snow}} = 0.003$  mwe PDD-1 and  $f_{PDD_{snow}}$  0.008 mwe PDD-1. Refreezing is considered, with a value of  $C_{si} = 60\%$ . GRISLI-UCM accounts for changes in elevation at each time step considering a linear atmospheric vertical profile for temperature with different lapse rates in summer and in the

annual mean (0.0080 and 0.0065 K m-1, respectively) to account for the smaller summer atmospheric vertical stability (see also the Supplementary Information).

Basal melting inland depends on pressure and water content at the base of the ice sheet **Rev. 2 SC 8** (Ritz et al., 2001) as well as on the geothermal heat flux, which is prescribed from **Rev. 2 SC 9** the reconstruction by Shapiro and Ritzwoller (2004).

5 Basal melting for floating ice is computed using a linear temperature anomaly with respect to the freezing point. The details of the implementation of the boundary conditions (SMB and oceanic basal melting) in this particular study are given below (Section 2.2).

**2.2 Offline forcing method**

10

SMB and oceanic basal melting are obtained through a time-varying synthetic climatology built through a novel method that is found to provide a more realistic offline forcing for ice-sheet models than classical offline methods (Banderas et al., 2017). The method follows a perturbative approach in the sense that the forcing combines the present-day climatology, obtained from observational data, together with simulated anomalies. But in contrast to usual offline forcing methods, orbital and millennial scale variabilities are not lumped in a sole anomaly pattern but differentiated. The method thus combines present-day ob-

15 simulated stadial-interstadial anomalies, scaled by a millennial-timescale index:

$$\boldsymbol{T}^{\text{atm}}(t) = \boldsymbol{T}_{0}^{\text{atm}} + (1 - \alpha^{\star}(t)) \boldsymbol{\Delta} \boldsymbol{T}_{\text{orb}}^{\text{atm}} + \beta^{\star}(t) \boldsymbol{\Delta} \boldsymbol{T}_{\text{mil}}^{\text{atm}}$$
(1)

servations, simulated Last Glacial Maximum (LGM) anomalies relative to present, scaled by an orbital-timescale index, and

$$\boldsymbol{P}(t) = \boldsymbol{P}_0 \left\{ \alpha^*(t) + (1 - \alpha^*(t)) \,\delta \boldsymbol{P}_{\rm orb} \left[ (1 - \beta^*(t)) + \beta^*(t) \,\delta \boldsymbol{P}_{\rm mil} \right] \right\}$$
(2)

Here, Tatm(t) and P(t) are the SAT and precipitation fields at time t. Tatm0 and P0 are the ERA-INTERIM present-day SAT and precipitation climatologies (Dee et al., 2011). ΔTatmorb = Tatmlgm - Tatmpd and δPorb = Plgm/Ppd are the orbital temperature
anomaly and precipitation ratio relative to the present day (not shown, see Banderas et al. (2017)), respectively, obtained from previous equilibrium simulations for the preindustrial and LGM climates performed with the CLIMBER-3α model (Montoya and Levermann, 2008). ΔTatmnil = Tatmis - Tatmst and δPmil = Pis/Pst are the millennial temperature anomaly and precipitation ratio, respectively, for the interstadial relative to the stadial state (Section 2.3). Rev. 1 SC 2: The key differences between these climate modes are that in the stadial, North Atlantic Deep Water (NADW) formation is relatively weak and takes place

- 25 south of Iceland. Accordingly the sea-ice front in the North Atlantic reaches 40°N. In the interstadial state there is a north-ward shift and intensification of NADW formation. Northward oceanic heat transport increases, and the North Atlantic and surrounding areas warm relative to the stadial state, in particular the Nordic Seas. The simulated interstadial state is thus characterised by a more vigorous NADW formation and AMOC together with reduced sea ice in the Nordic Seas, and a temperature increase of up to 10 K in the North Atlantic relative to the stadial state, with a maximum anomaly in the Nordic Seas. Note
- 30 bold symbols indicate two-dimensional spatial fields. The stadial mode in our study is represented by a climate simulation of the LGM with CLIMBER-3 $\alpha$  (Montoya and Levermann, 2008). The interstadial mode is taken from a recent glacial transient simulation performed with the same model under glacial climatic conditions, but with intensified NADW formation (Banderas et al., 2015).  $\alpha^*$  and  $\beta^*$  are two indices that separately modulate the contribution of the orbital and millennial anomalies. Both

were built based on two recent complementary temperature reconstructions over Greenland, one from the NGRIP ice-core record for the LGP (Kindler et al., 2014), and the other one from several ice-core records for the Holocene (Vinther et al., 2009). Their combination (hereafter, the KV reconstruction) results in a continuous temperature reconstruction for Greenland for the past 120 ka (Banderas et al., 2017).  $\alpha^*$  is obtained after applying a low-pass frequency filter ( $f_c = 1/18 \text{ ka}^{-1}$ ) to the

5 original KV reconstruction based on a spectral decomposition;  $\beta^*$  is obtained following a similar procedure but retaining the high frequency signal. Both indices are tuned in such a way that the resulting synthetic temperature time series at the NGRIP site exactly matches the KV reconstruction (this distinguishes  $\alpha^*$  and  $\beta^*$  from the raw  $\alpha$  and  $\beta$  indices previous to this tuning; Banderas et al. (2017)).

The net basal melting rate for floating parts  $\boldsymbol{B}$  is assumed to follow a linear relation:

10
$$\boldsymbol{B} = \kappa \left( \boldsymbol{T}^{\text{ocn}} - \boldsymbol{T}_{f} \right)$$
 (3)

where  $T^{\text{ocn}}$  is the oceanic temperature close to the grounding line,  $T_f$  is the temperature at the ice base, assumed to be at the freezing point, and  $\kappa$  is the heat flux exchange coefficient between ocean water and ice at the ice-ocean interface. **Rev. 2 SC 15, 29 and 32** Several marine-shelf basal melting parameterizations can be found in the literature. The submarine melt rate is thought to be directly influenced by the oceanic temperature variations below the ice shelves. Accordingly, most basal melting

- 15 parameterizations are built as a function of the difference between the oceanic temperature at the ice-ocean boundary layer and the temperature at the ice-shelf base, generally assumed to be at the freezing point. The dependence on this temperature difference can be linear (Beckmann and Goosse, 2003) or quadratic (Holland et al., 2008; Pollard and DeConto, 2012; DeConto and Pollard, 2016; Pattyn, 2017). The linear marine-shelf basal melting parameterization used in this study is the simplest case that allows testing of the ice-sheet sensitivity to past oceanic temperature changes. Nevertheless, it accounts separately for
- 20 sub-ice-shelf areas near the grounding line and for purely floating ice (ice shelves). The basal melting rate for purely floating ice shelves (Bsh) is given by the grounding-line basal melt Bgl scaled by a constant factor

$$B_{\rm sh} = \gamma B_{\rm gl}(t) \tag{4}$$

In this study,  $\gamma$  is set to 0.1. Thus, we consider that the submarine melting rate for ice shelves is 10 times lower than that close to the grounding zone, which is qualitatively in agreement with observations in some Greenland glaciers (Münchow et al.,

25 2014; Rignot and Jacobs, 2002; Wilson et al., 2017). The melt rate in the open ocean, that is considered as being beyond the continental shelf break, is prescribed to a high value (20 m  $a^{-1}$ ) to avoid unrealistic ice growth beyond 750 m of ocean depth, following Peyaud et al. (2007).

Following the approach described above,  $T^{ocn}(t)$  is assumed to be given by an expression analogous to Eq. 1. Thus Eq. 3 can be rewritten as:

30
$$\boldsymbol{B} = \boldsymbol{B}_0 + \kappa \left[ (1 - \alpha^*(t)) \, \boldsymbol{\Delta} \boldsymbol{T}_{\text{orb}}^{\text{ocn}} + \beta^*(t) \, \boldsymbol{\Delta} \boldsymbol{T}_{\text{mil}}^{\text{ocn}} \right]$$
(5)

where  $B_0 = \kappa (T_0^{\text{ocn}} - T_f)$  represents the present-day oceanic basal melting rate.

**Rev. 2 GC4, SC2 and SC17:** Finally, millennial-scale sea-level variations are prescribed according to the reconstruction by Grant et al. (2012, Section 2.3). The specific details of the experimental setup used are described below.

**2.3 Experimental setup**

25

We herein investigate the response of the EIS to millennial-scale climate variability during MIS 3. The starting point of our experiments is a control-run ice-sheet simulation with constant bounday conditions for MIS 3 that provides a representative configuration of the EIS for that time period (Figure 1). To this end,  $\alpha^*$  was set to its value at 40 ka BP, that is,  $\alpha^* = \alpha_{40K}^* = \alpha_{40K}^*$

5 -0.1, and  $\beta^* = 0$  to preclude millennial-scale variations. Note however these values are to a certain extent arbitrary; they are intended to provide a stable mean background state similar but not neccessarily identical to background MIS 3 conditions. Thus:

$$\boldsymbol{T}_{40K}^{\text{atm}} = \boldsymbol{T}_{0}^{\text{atm}} + (1 - \alpha_{40K}^{\star}) \boldsymbol{\Delta} \boldsymbol{T}_{\text{orb}}^{\text{atm}}$$
(6)

$$\boldsymbol{P}_{40K} = \boldsymbol{P}_0 \left[ \alpha_{40K}^{\star} + (1 - \alpha_{40K}^{\star}) \delta \boldsymbol{P}_{\text{orb}} \right]$$
(7)

10
$$\boldsymbol{B}_{40K} = \boldsymbol{B}_0 + \kappa (1 - \alpha_{40K}^{\star}) \boldsymbol{\Delta} \boldsymbol{T}_{\text{orb}}^{\text{ocn}}$$
(8)

Note that although Eq. 8 is formally correct and consistent with the scheme used, in contrast to the present-day SAT or precipitation the present-day rate of oceanic basal melting cannot be determined. Thus, in practice we replace this equation by directly tuning the value of  $B_{40K}$  to obtain a reasonable ice-sheet configuration at 40 ka BP given the atmospheric forcing fields expressed by equations 6-7. To this end, a constant basal melting rate of 0.1 m a-1 is assumed. **Rev. 1 SC 3**: The ice

- 15 sheet was forced with the resulting climatologies for 100 kyr previous to the starting of the perturbations described below. This allows the vertical temperature profile within the ice sheet to be equilibrated with the climate. This procedure was found to facilitate the growth of European ice-sheets to an extent that satisfactorily agrees with previous reconstructions (Svendsen et al., 2004; Kleman et al., 2013).
- Our forcing method allows to investigate the response of the EIS solely to millennial-scale climate variability at MIS 3 by 20 keeping constant the orbital component of the forcing ( $\alpha^* = \alpha^*_{40K}$ ) and letting  $\beta^*$  vary throughout the LGP (eqs. 1, 2 and 5). In order to assess the relative roles of the atmosphere and the ocean, three independent experiments have been carried out. First, an atmospheric-only forced simulation (ATM) in which the time evolution of SAT and precipitation on millennial time scales is considered, while the oceanic forcing is kept constant to MIS 3 (i.e., 40 ka BP) background climatic conditions. Thus:

$$\boldsymbol{T}^{\text{atm}}(t) = \boldsymbol{T}^{\text{atm}}_{40K} + \beta^{\star}(t) \, \boldsymbol{\Delta} \boldsymbol{T}^{\text{atm}}_{\text{mil}} \tag{9}$$

$$\boldsymbol{P}(t) = \boldsymbol{P}_{40K} \left[ (1 - \beta^{\star}(t)) + \beta^{\star}(t) \,\delta \boldsymbol{P}_{\text{mil}} \right] \tag{10}$$

$$\boldsymbol{B}(t) = \boldsymbol{B}_{40K} \tag{11}$$

Second, an oceanic-only forced simulation OCN in which the atmospheric forcing is kept constant while the oceanic basal melting is allowed to vary at millennial timescales around its background MIS 3 value:

$$\boldsymbol{T}^{\text{atm}}(t) = \boldsymbol{T}^{\text{atm}}_{40K} \tag{12}$$

$$\mathbf{30} \qquad \mathbf{P}(t) = \mathbf{P}_{40K} \tag{13}$$

$$\boldsymbol{B}(t) = \boldsymbol{B}_{40K} + \kappa \beta^{\star}(t) \, \boldsymbol{\Delta T}_{\text{mil}}^{\text{ocn}} \tag{14}$$

The magnitude and sign of oceanic temperature anomalies  $\Delta T^{\text{ocn}}$  depends on the depth at which  $T^{\text{ocn}}$  is considered. In our simulations, a large part of the NE sector of the EIS **Rev 1 GC4 and Rev 2 GC9:** is marine based with shallow bedrock depths between 500 m and less than 100 m in several locations further south. It is therefore unknown whether this marine ice sheet was more susceptible to changes in the surface or the subsurface of the ocean. To investigate the effect of this uncertainty,

5 we decided to perform two different simulations considering different depths: one corresponding to the surface (OCNsrf) and the other one considering deeper (subsurface) oceanic waters by averaging temperatures within the range of 400-600 m depth (OCNsub). Therefore we hereafter distinguish between  $\Delta T_{mil}^{ocn}$  for surface or subsurface millennial-scale temperature anomalies, respectively (Figure 2). The realism and convenience of aplying one or the other is adressed in section 5. Finally, a simulation ALL combining both the atmospheric and the oceanic forcings:

10
$$T^{\text{atm}}(t) = T^{\text{atm}}_{40K} + \beta^{\star}(t) \Delta T^{\text{atm}}_{\text{mil}}$$
 (15)

$$\boldsymbol{P}(t) = \boldsymbol{P}_{40K} \left[ (1 - \beta^{\star}(t)) + \beta^{\star}(t) \,\delta \boldsymbol{P}_{\text{mil}} \right] \tag{16}$$

$$\boldsymbol{B}(t) = \boldsymbol{B}_{40K} + \kappa \beta^{\star}(t) \, \boldsymbol{\Delta T}_{\text{mil}}^{\text{ocn}} \tag{17}$$

In all experiments  $\beta^{\star}(t)$  dictates the millennial-scale variability of the forcings (Figure 3). Because our simulated stadialto-interstadial transition results from an intensification of the AMOC, positive  $\beta^{\star}$  values imply an increase in  $T^{\text{atm}}$  relative

to its background MIS 3 value (e.g., Eq. 15 and Figures 2 and 3). As a consequence, the atmosphere warms at interstadials relative to stadial periods, as reflected by the  $\Delta T_{mil}^{atm}$  millennial-scale anomaly field (Figure 2). Rev 2 GC1: Note that for  $\kappa \beta^* \Delta T_{mil}^{ocn} < -B_{40K}$  refreezing is allowed (B(t) < 0). Following Obase et al. (2017) in order to avoid unrealistic values of ice accretion under the ice shelves the sensitivity to the ocean is decreased by one order of magnitude when basal melting becomes negative, i.e., the refreezing rate is reduced to 10% of its estimated value following Eq. 5.:

20
$$\kappa = \begin{cases} \kappa_0 & \text{if } \boldsymbol{B}(t) > 0\\ 0.1 \cdot \kappa_0 & \text{if } \boldsymbol{B}(t) < 0 \end{cases}$$
(18)

where  $\kappa_0$  is the nominal value of the heat exchange coefficient. An ensemble of simulations for different values of  $\kappa_0$  have been considered to evaluate the sensitivity of the EIS to the forcing. **Rev. 2 GC4, SC2 and SC17:** Finally, varying sea-level forcing is considered (Figure 3b), both alone (SL run) and in combination with the previous forcings (ATM+OCN+SL).

**3** Results**

- 25 Substantial differences are found in the response of the EIS to the forcing scenarios. **Rev. 2 GC4, SC2 and SC17:** Under constant forcing, the CTRL run shows negligible millennial-scale sea-level equivalent (SLE) variations, although a lower frequency SLE fluctuation is found related to internal ice-sheet variability (Figure 3). **Rev. 1 SC17, Rev. 2 GC4:** When the model is forced only by changes in sea level (SL run), a slight response is observed on millennial-scales. These changes appear not be sufficient to cause a substantial migration of the grounding line, thus not affecting ice velocities. In ATM, the
- 30 atmospheric forcing alone causes a sequence of enhanced ablation episodes resulting in modest ice volume variations (up to

1.5 m SLE) during the most prominent stadial-interstadial transitions. In contrast, **Rev. 1 GC4 and Rev 2 GC9:** the oceanic forcing in OCNsrf induces pronounced changes in the dynamics of the EIS on millennial time scales, with episodes of a large volume reduction occurring during interstadials. The combination of sea level, atmospheric and oceanic forcings (SL + ATM + OCNsrf run) results in a similar response of the EIS to that obtained in OCNsrf (Figure 3) as a consequence of the larger effect

- 5 of the oceanic forcing in OCNsrf with respect to ATM. OCNsub shows an anti-phase relationship with respect to OCNsrf, with the largest reductions in ice volume occurring during prolonged stadial periods and regrowth phases happening during interstadials. This behavior can be explained by the fact that ocean waters at the subsurface warm (cool) during episodes of reduced (enhanced) convection at the Nordic Seas as a result of variations in the AMOC strength. Thus, the out-of-phase relationship found in the dynamic response of the EIS among these two oceanic experiments relates to the opposed sign of their
- 10 spatial forcing patterns (Figure 2). When considering the forcing at the subsurface of the ocean together with the atmosphere (SL+ATM+OCNsub), slight reductions of the EIS volume (less than 1 m of s.l.e) during interstadials are superimposed onto the previous behavior (Figure 3).

**Rev2 GC1:** The magnitude of these changes for the MIS 3 period is illustrated in Figure 4. The simulation forced with the surface of the ocean (OCNsrf) and that including the rest of the forcings (OCNsrf + ATM + SL) show the largest amplitudes,

- 15 with peaks of sea-level rise above 4mm yr-1 during DO-events and sustained contributions well above 1 mm yr-1 during entire interstadial periods. In ATM, a decline of the EIS during stadial-to-interstadial transitions is still observed but presents a smaller amplitude of 1-2 mm yr-1. The simulations in which the ice sheet is forced with the subsurface of the ocean present a decline of their volume during stadial periods and regrowth during interstadials as a consequence of the inverted spatial pattern of temperature anomalies with respect to the surface. In the case of OCNsub the amplitude of these changes is smaller than
- 20 in the OCNsrf case, on the order of 0.5-1 mm yr-1, and reaches more than 1 mm yr-1 during pronounced stadials (as ca. at 44 ka BP). The OCNsub + ATM + SL simulation shows a slight volume loss during interstadials, as a consequence of the atmospheric forcing, that is superimposed onto the OCNsub behaviour.

**Rev 1 GC 5:** The response of the EIS has been analyzed in terms of its mass balance decomposition for the all-forcing runs (Figure 5). The surface ocean temperature varies in phase with the atmosphere. Thus, during stadial-to-interstadial transitions

- 25 the high negative values of dV/dt can be explained by the conjunction of an initial sharp increase in ablation together with pronounced increases in basal melting and calving, which allow a large grounding line retreat in the Bjørnøyrenna basin (Figure 5 mid panel). The rate of ice loss by basal melting is similar to that resulting from the increase in ablation (as reflected in the surface mass balance) during the peak of a stadial-to-interstadial period. However, basal melting is much more efficient than surface mass balance in decreasing volume along the whole duration of an interstadial. This is due to the fact that ablation
  - 30 is restricted to the southern borders of the EIS. Thus, when the ice sheet has retreated to areas of no ablation, in spite of a slight further loss provided by the elevation feedback it rapidly equilibrates and a negative surface mass balance can not propagate further inland. In contrast, when enhanced basal melting from higher oceanic temperatures is applied, the associated retreat can propagate further inland occupying a large proportion of the Bjørnøyrenna basin and facilitating high rates of volume loss (although similar in amplitude with respect to SMB) during the whole interstadial period (see the animation in the
  - 35 Supplementary Information). Rev. 2 SC 16: During stadial periods, both enhanced positive mass balance and negative oceanic

anomalies (producing a slight refreezing) favor the regrowth of the EIS. Subsurface ocean temperatures evolve also in phase with the atmosphere in the SW part of the EIS but in anti-phase in its NE part. In other words, when forcing with the subsurface of the ocean, a slight warming (cooling) is observed around the Britain ice sheet while cooling (warming) of the Bjørnøyrenna basin is simulated during interstadial (stadial) periods (see Figure 2). Therefore, the OCNsub + ATM + SL simulation presents

- 5 volume declines during stadial-to-interstadial transitions due to an increase in ablation and basal melting in the SW part. Rev. 2 SC 16: Subsequently, reduced basal melting in the NE part of the EIS (allowing even a slight refreezing at its grounding line) favors regrowth of the Bjørnøyrenna basin during interstadial periods. Finally, shifting to pronounced stadial periods (as in ca. 44 ka BP) favors the penetration of warm subsurface waters that increase basal melting enough to produce an ice-sheet retreat in the NE part in spite of the enhanced positive surface mass balance. When considering the atmosphere and the subsurface ocean
- 10 forcing together, these competing processes translate into a smaller amplitude of millennial-scale EIS changes as compared to the case with surface ocean forcing. Furthermore, declines of the EIS can be observed both during the beginning of interstadial periods and during pronounced stadial periods in OCNsub + ATM + SL.

**Rev1 GC4 and Rev2 GC 9:** Focusing on the OCN and ATM simulations separately facilitates isolating the effects of the ocean on this complex pattern. To this end, the simulated ice-sheet distribution and velocities of OCNsrf, OCNsub and ATM

- 15 are shown in Figure 6 for the period around DO-event 12, at ca. 47 ka BP. As expected, OCNsrf shows a widespread retreat both in the NE and the SW of the EIS from the stadial to the interstadial period. This is accompanied by an acceleration of the Bjørnøyrenna basin due to its grounding line thinning and retreat (Figure 6, left panels). OCNsub presents a collapsed Bjørnøyrenna basin during the stadial period previous to DO-event 12 due to enhanced basal melting from warmer subsurface waters. The transition to the interstadial period favors the regrowth of this NE part of the EIS due to decreased basal melting,
- 20 while its SW section slightly retreats (Figure 6 mid panels). Concerning ATM, only in the southwestern (SW) part of the EIS is the atmospheric forcing capable of generating an important reduction in the EIS volume in response to the stadial-interstadial transition (Figure 6 right panels). This is a result of the spatial pattern of the forcing, with the largest SAT anomalies located around the Nordic seas (Figure 2). Therefore, the ice volume reduction of the EIS in ATM is due to the positive SAT anomaly, which leads to enhanced ablation in the SW part of the EIS. In turn, reduced SATs during stadials allow the regrowth of the ice
- 25 sheet up to the continental margin of the Nordic seas. The more active dynamic response of the EIS in OCN simulations can be attributed to the increase in oceanic temperatures by 2-4°C (Figure 2) within the margins of the ice-sheet during interstadial (in the case of OCNsrf) and stadial (OCNsub case) periods, which translates into enhanced basal melting at the margins of the EIS. The SW sector of the EIS also responds to the warmer SSTs, actually with a larger reduction of ice volume than in ATM (Figure 6).
- 30 The spatial patterns shown in Figure 6 are representative of the stadial-to-interstadial transitions. **Rev1 GC4 and Rev1 SC9:** In OCNsrf, the EIS reacts to every abrupt surface warming with a substantial ice-flow acceleration, especially in the Bjørnøyrenna basin (Figure 7). Ice shelves that are present during stadial periods suddenly retreat during DO-events and together with enhanced basal melting favor thinning and retreat of the grounding line that translate in large iceberg discharges up to ca. 0.06 Sv. In the OCNsub case, ice velocities in the Bjørnøyrenna basin increase during stadials, when enhanced basal
- 35 melting erodes the grounding line and favors its retreat. Peaks in calving are recorded accordingly during pronounced stadial

periods. These peaks are however of less amplitude than in the OCNsrf case. This can by explained by the fact that transitions to stadials are usually more gradual than transitions to interstadials, thus the incursion of warmer waters happens in this case in a smoother manner. High velocities reach their maxima at the end of the stadial and beginning of the interstadials. The latter are however not accompanied by an increase in calving due to the fact that ice shelves are expanding and thickening

5 during this period thanks to reduced basal melting (Figure 8). In general, the extension of ice shelves is greatly reduced during periods of enhanced basal melting (Figures 7, 8), with no large unconfined ice shelves surviving during these episodes (see the Supplementary Information). Some thinner ice shelves remain, in spite of the enhanced basal melting, thanks to an increase in advection from the Bjørnøyrenna ice stream triggered by a grounding line retreat (Figure 6).

Rev. 1 SC 11: Note that changes in the position of the calving front are usually accompanied by a grounding line displacement. For some minor ice-shelf breakups this close relationship can be broken, but with almost no effects upstream inland. Thus we consider that the grounding line position is the best indicator for characterizing the dynamic behavior of the marine part of the EIS.

Inspection of the temporal evolution of the grounding line position in OCN simulations confirms that ice dynamics control the majority of ice-volume variations in the EIS as opposed to the SMB processes involved in ATM. The migration of the

15 grounding line through time has been characterized by means of an index ( $\mu$ ) that weighs the proportion of non-grounded points in the region of the Bjørnøyrenna basin:

$$\mu(t) = \left(1 - \frac{N_g(t)}{N}\right) \cdot 100\tag{19}$$

where  $N_g(t)$  represents the evolution of the number of points of grounded ice within a fixed area of N points in the Barents Sea region. While in ATM  $\mu$  barely changes (Figure 9), OCN runs show a large dynamic behavior of the basin. **Rev1 GC4**

- **20** and Rev1 SC9: In OCNsrf,  $\mu$  reflects a synchronous evolution of the grounding line position and the oceanic forcing, with major retreats coinciding with interstadial states (Figure 9). Conversely, the Bjørnøyrenna basin is generally much closer to a full retreat in OCNsub due to a larger penetration of warm subsurface waters (Figure 2,right compared to 2,middle) during stadials. However, the grounding line is able to advance and reach Svalbard during episodes of reduced basal melting at the interstadials (Figure 9).
- The direct coupling between the oceanic forcing and the response of the Bjørnøyrenna ice stream is also evident from the relatively high negative correlation ( $r \simeq -0.70$ ) found between  $\mu$  and ice thickness (Figure 9). In essence, in response to the grounding-line retreat (advance), acceleration (deceleration) of the flow takes place upstream in the Bjørnøyrenna ice stream, as reflected by the nearly linear positive correlation ( $r \simeq 0.93$ ) found between  $\mu$  and velocities in the channel (Figure 9). (Figure 7).

**30 The only thing left: ADD OCNsub in figure 9, and describe it.**

**Rev1 GC1 and Rev2 SC1:** As a consequence of the destabilization of the ice sheet, important ice-volume variations are observed in the NE part of the EIS during millennial-scale climatic transitions, which added to the minor contribution of the SW retreat, result in fluctuations of more than 4 m SLE in OCNsrf, up to 2.5 m in OCNsub and ca. 1 m in ATM (Figure 3).

In order to investigate the sensitivity of the results to the model parameters, eight additional OCN simulations, both for the surface and the subsurface, have been carried out with different  $\kappa_0$  parameters between 1-9  $m a^{-1} K^{-1}$ , i.e., bracketing our standard case of  $\kappa_0 = 5 m a^{-1} K^{-1}$ . This choice reflects the inferences based on measurements made on Antarctic ice shelves that a variation of 1 K in the effective oceanic temperature changes the melt rate by ca. 10  $m a^{-1}$  (Rignot and Jacobs,

5 2002; Shepherd et al., 2004). A robust response of the EIS is found, with a more reactive EIS response for increasing  $\kappa$  values (see Supplementary Information). The sensitivity of our results to the values of the atmospheric mass balance model has also been explored. In spite of largely exploring the values of the parameters that determine the sensitivity to surface mass balance, the EIS variability induced by the ocean is always found to be of greater amplitude than the one induced by the atmosphere provided that  $\kappa_0 > 3 m a^{-1} K^{-1}$  (see Supplementary Information).

**10 4 Discussion**

Our results suggest a highly dynamic Eurasian ice sheet at millennial time-scales largely responding to changes in the ocean temperatures. **Rev. 2 GC 4:** Some authors present the marine based Kara-Barents complex as an analogue for present-day West Antarctic ice sheet for which bedrock topography is a major control for stability. Marine ice sheet instability is generally triggered by a sub-shelf basal melt perturbation but is largely amplified by the local bedrock depth. We have shown, in this

15 sense, that the Bjørnøyrenna basin is highly susceptible to changes in the oceanic temperatures. The timing of this response with respect to changes registered in Greenland depends, however, on whether the surface or the subsurface of the ocean is considered as the relevant forcing of the ice sheet.

Recently, IRD peaks of Fennoscandian origin reported from a high-resolution marine sediment core from the Norwegian Sea indicate the presence of more frequent IRD deposition and thus calving during interstadials than during stadials (Dokken et al.,

- 20 2013). This result has been corroborated in a compilation of new and previously published data (Becker et al., 2017) clearly showing that within MIS 3, the IRD deposition increases within interstadials. The coeval deposition of carbonate-rich, sorted fine sands and near-surface warming suggests the presence of Atlantic water along the margin, and is interpreted by the authors as the effects of winnowing due to an intensified AMOC during interstadials. This interpretation results in concordance with our results when considering the surface waters as the oceanic forcing. Thus, this agreement would play in favor of considering
- 25 that the EIS was primarily responding to changes in the surface of the ocean.

Our results also provide a mechanism to explain the pervasive presence of IRD in the North Atlantic during MIS 3, both during stadials and interstadials, and originating both in the LIS and the EIS. During stadials, the simultaneous appearance of IRD across the wider North Atlantic Ocean can be explained through the build-up of subsurface heat in the high-latitude North Atlantic leading to increased iceberg calving in the presence of large, thick ice shelves, together with lower surface

30 temperatures allowing for wider dispersal of icebergs (Barker et al., 2015). According to our results interstadials could lead to enhanced calving of the EIS through oceanic surface subglacial melting as a result of the warmer surface conditions and relatively shallow grounding line of this ice sheet. The identification of IRD layers with increased calving through ice-sheet instabilities must be taken with caution, since it is based on several untested assumptions (Clark and Pisias, 2000): (i) delivery of IRD to a specific site is caused solely by iceberg calving, versus transport by sea-ice; (ii) an increase in IRD represents an increase in the iceberg flux, versus a greater amount of debris incorporated at the base of the ice sheet that delivers the icebergs, or a greater distance of iceberg transport; (iii)

- 5 the amount of IRD carried by all the icebergs is similar, assuming therefore a direct relationship between IRD concentration and iceberg flux. However, the former assumptions have not been confirmed and, thus, the calving-IRD relationship might not be so direct. In addition, ocean temperatures affect melting of icebergs and thus their release of IRD. Variations in ocean temperatures can alter the IRD released by an iceberg at a certain site, causing variations in IRD deposition even for a constant amount of icebergs produced at the source.
- 10 Our experimental setup is not intended to match the paleorecord, but to provide insight into the response of the EIS to millennial-scale variability. The EIS variations simulated here represent the upper-end amplitude of potential responses during the whole glacial cycle, due to its large size. Extending the study to cover the whole LGP would require the consideration of orbital variability as part of the forcing. In this case, the EIS would likely be smaller during the mildest phase of MIS 3, thus limiting its contact with the ocean and the production of iceberg discharges.
- 15 Also, our results depend somewhat on the particular SAT and **Rev 1 GC 4:** oceanic temperature anomaly patterns simulated by our climate model, the magnitudes of the resulting forcing, and the initial size of the simulated EIS. **Rev 1 GC 1:** The use of different atmospheric realisations is subject to the availability of climate simulations with different models for the three climate states needed: glacial (stadial), present, and interstadial. The latter is only available for a reduced number of models. This makes the assessment of this issue difficult in the present study. Assessing the sensitivity to these features should be in
- 20 the scope of future work, and illustrates the need for carrying out new simulations of both the interstadial and the stadial states with more sophisticated climate models. Rev. 2 SC1: Nonetheless, our results indicate that the ocean is the major driver of the EIS ice-volume changes during MIS-3. Note the temporal index used is the same for the atmosphere and the ocean and the amplitude is given by an OGCM simulation of two different oceanic states mimicking stadial and interstadial periods. We then translate those fields into ablation (through PDD, whose uncertainty has been extensively explored) and into basal melting
- 25 (through a linear equation). The values of the oceanic sensitivity parameter ( $\kappa$ ) we used here are in the range (or even below in most cases) of those suggested by data in Antarctica (Rignot et al. 2002). Note, in particular, that even for low-mid values of  $\kappa$  of 3 meter/year/Kelvin the response to the ocean appears to be of greater amplitude than that to the atmosphere, making our main conclusions robust Finally, our study lacks bi-directional coupling between the ice sheet, the atmosphere and the ocean. Eventually the goal is to investigate this matter with fully coupled climate-ice sheet models.
- 30 Meltwater discharge from the EIS and other ice sheets surrounding the Nordic Seas is often implied as a cause of ocean instabilities. The same would be the case for iceberg discharges. This issue is beyond the scope of this study; its assessment would require investigating the impact of these freshwater perturbations in deep water formation and the AMOC. Again, proper assessment requires the use of a coupled climate-ice sheet model.

**5 Conclusions**

We have investigated the response of the EIS to millennial-scale climate variability associated with DO-events through a series of simulations with a three-dimensional, hybrid ice-sheet model that represents inland ice flow under the SIA and floating ice shelves and ice streams through the SSA. The model makes use of an offline forcing method that separately accounts for

orbital and millennial-scale climate variability during the LGP, improving the representation of the latter (Banderas et al., 2017).
 Atmospheric and ocean forcings associated with millennial-scale variability were considered both separately and together.
 Separating the effects of atmospheric and oceanic forcing during the glacial period has allowed us to quantify the contribution

of each to EIS variability. Atmospheric forcing during stadial-interstadial transitions has a **Rev2 SC1**:very modest effect on the ice sheet, which is a consequence of the largest SMB changes being confined to SW sector of the EIS, where the forcing

- 10 is strongest. In contrast, the oceanic forcing has a Rev2 SC1: much larger effect, through changes in the ice dynamics in the Bjørnøyrenna basin of the EIS. Ocean warming is able to induce a retreat of grounded ice in this part of the EIS through dynamic processes. As a consequence, significant ice-volume variations result during millennial-scale climatic transitions. Added to the Rev2 SC 18: smaller contribution of the SW retreat, this results in sea-level changes on the order of several meters. Sensitivity experiments for different values of the oceanic heat coefficient parameter show that this is a robust response
- 15 of the model.

Our results thus support the existence of a highly dynamic EIS during the LGP. They suggest an important role of oceanic melt forcing through changes in the ocean circulation in controlling the ice-stream activity. Together with previous work (Alvarez-Solas et al., 2013), they imply that oceanic circulation changes and the associated ocean-ice sheet interactions are able to explain virtually all ice rafting events in the North Atlantic within MIS 3, from the H-events of the LIS during stadials

20 to those of the EIS during interstadials. Additionally, our results highlight the need for stronger constraints on the local North Atlantic behavior in order to shed light on the ice sheet's glacial dynamics.

Competing interests. The authors declare no competing interests

[revised manuscript text omitted]

- 10 in the North Atlantic, Paleoceanography, 21, 2006.
  - Montoya, M. and Levermann, A.: Surface wind-stress threshold for glacial Atlantic overturning, Geophys. Res. Lett., 35, L03608, https://doi.org/10.1029/2007GL032560, 2008.
  - Münchow, A., Padman, L., and Fricker, H. A.: Interannual changes of the floating ice shelf of Petermann Gletscher, North Greenland, from 2000 to 2012, Journal of Glaciology, 60, 489–499, 2014.
- 15 Obase, T., Abe-Ouchi, A., Kusahara, K., Hasumi, H., and Ohgaito, R.: Responses of Basal Melting of Antarctic Ice Shelves to the Climatic Forcing of the Last Glacial Maximum and CO2 Doubling, Journal of Climate, 30, 3473–3497, 2017.

Olsen, L., Sveian, H., Borg, K., Bergstram, B., and Broekmans, M.: Rapid and rhythmic ice sheet fluctuations in western Scandinavia 15-40 Kya–a review, Polar Research, 21, 235–242, 2002.

- Patton, H., Hubbard, A., Andreassen, K., Winsborrow, M., and Stroeven, A. P.: The build-up, configuration, and dynamical sensitivity of the
  Eurasian ice-sheet complex to Late Weichselian climatic and oceanic forcing, Ouaternary Science Reviews, 153, 97–121, 2016.
- Pattyn, F.: Sea-level response to melting of Antarctic ice shelves on multi-centennial timescales with the fast Elementary Thermomechanical Ice Sheet model (f. ETISh v1. 0), The Cryosphere, 11, 1851, 2017.
  - Peck, V., Hall, I., Zahn, R., Grousset, F., Hemming, S., and Scourse, J.: The relationship of Heinrich events and their European precursors over the past 60ka BP: a multi-proxy ice-rafted debris provenance study in the North East Atlantic, Quaternary Science Reviews, 26, 862–875–2007
- 25 862–875, 2007.
  - Peyaud, V., Ritz, C., and Krinner, G.: Modelling the Early Weichselian Eurasian Ice Sheets: role of ice shelves and influence of ice-dammed lakes, Climate of the Past Discussions, 3, 221–247, 2007.
  - Pollard, D. and DeConto, R.: Description of a hybrid ice sheet-shelf model, and application to Antarctica, Geoscientific Model Development, 5, 1273, 2012.
- 30 Rasmussen, T. L. and Thomsen, E.: Pink marine sediments reveal rapid ice melt and Arctic meltwater discharge during Dansgaard-Oeschger warmings, Nature communications, 4, 2849, 2013.

Reeh, N.: Parameterization of melt rate and surface temperature on the Greenland ice sheet, Polarforschung, 59, 113–128, 1989.
Rignot, E. and Jacobs, S. S.: Rapid bottom melting widespread near Antarctic ice sheet grounding lines, Science, 296, 2020–2023, 2002.
Ritz, C., Rommelaere, V., and Dumas, C.: Modeling the evolution of Antarctic ice sheet over the last 420,000 years: Implications for altitude

- changes in the Vostok region, Journal of Geophysical Research: Atmospheres (1984–2012), 106, 31943–31964, 2001.
  - Scourse, J. D., Hall, I. R., McCave, I. N., Young, J. R., and Sugdon, C.: The origin of Heinrich layers: evidence from H2 for European precursor events, Earth and Planetary Science Letters, 182, 187–195, 2000.

- Scourse, J. D., Haapaniemi, A. I., Colmenero-Hidalgo, E., Peck, V. L., Hall, I. R., Austin, W. E., Knutz, P. C., and Zahn, R.: Growth, dynamics and deglaciation of the last British–Irish ice sheet: the deep-sea ice-rafted detritus record, Quaternary Science Reviews, 28, 3066–3084, 2009.
- Shapiro, N. M. and Ritzwoller, M. H.: Inferring surface heat flux distributions guided by a global seismic model: particular application to
   Antarctica, Earth and Planetary Science Letters, 223, 213–224, 2004.
- Shepherd, A., Wingham, D., and Rignot, E.: Warm ocean is eroding West Antarctic ice sheet, Geophysical Research Letters, 31, 2004.
- Siegert, M. J. and Dowdeswell, J. A.: Numerical reconstructions of the Eurasian Ice Sheet and climate during the Late Weichselian, Quaternary Science Reviews, 23, 1273–1283, 2004.
- Steffensen, J., Andersen, K., Bigler, M., Clausen, H., Dahl-Jensen, D., Fischer, H., Goto-Azuma, K., Hansson, M., Johnsen, S., Jouzel, J.,
- 10 Masson-Delmotte, V., Popp, T., Rasmussen, S., Röthlisberger, R., Ruth, U., Stauffer, B., Siggaard-Andersen, M., Sveinbjörnsdóttir, A., Svensson, A., and White, J.: High-Resolution Greenland Ice Core Data Show Abrupt Climate Change Happens in Few Years, Science, 321, 680–684, doi: 10.1126/science.1157707, 2008.
  - Svendsen, J. I., Alexanderson, H., Astakhov, V. I., Demidov, I., Dowdeswell, J. A., Funder, S., Gataullin, V., Henriksen, M., Hjort, C., Houmark-Nielsen, M., et al.: Late Quaternary ice sheet history of northern Eurasia, Quaternary Science Reviews, 23, 1229–1271, 2004.
- 15 Vinther, B. M., Buchardt, S. L., Clausen, H. B., Dahl-Jensen, D., Johnsen, S. J., Fisher, D., Koerner, R., Raynaud, D., Lipenkov, V., Andersen, K., et al.: Holocene thinning of the Greenland ice sheet, Nature, 461, 385–388, 2009.
  - von Kreveld, S. v., Sarnthein, M., Erlenkeuser, H., Grootes, P., Jung, S., Nadeau, M., Pflaumann, U., and Voelker, A.: Potential links between surging ice sheets, circulation changes, and the Dansgaard-Oeschger cycles in the Irminger Sea, 60–18 kyr, Paleoceanography, 15, 425– 442, 2000.
- 20 Wilson, N., Straneo, F., and Heimbach, P.: Satellite-derived submarine melt rates and mass balance (2011–2015) for Greenland's largest remaining ice tongues, The Cryosphere, 11, 2773, 2017.

Wohlfarth, B.: Ice-free conditions in Sweden during Marine Oxygen Isotope Stage 3?, Boreas, 39, 377-398, 2010.